# SAC Flow: Sample-Efficient Reinforcement Learning of Flow-Based Policies via Velocity-Reparameterized Sequential Modeling

**Yixian Zhang[1][*], Shu'ang Yu[2,4][*], Tonghe Zhang[3], Mo Guang[4], Haojia Hui[4], Kaiwen Long[4], Yu Wang[2], Chao Yu[1][†], Wenbo Ding[1][†]**

[1]SIGS, Tsinghua University    [2]EE, Tsinghua University    [3]Carnegie Mellon University
[4]Li Auto    [5]Shanghai AI Laboratory

[*]Equal contribution    [†]Corresponding Authors
{yuchao,ding.wenbo}@sz.tsinghua.edu.cn

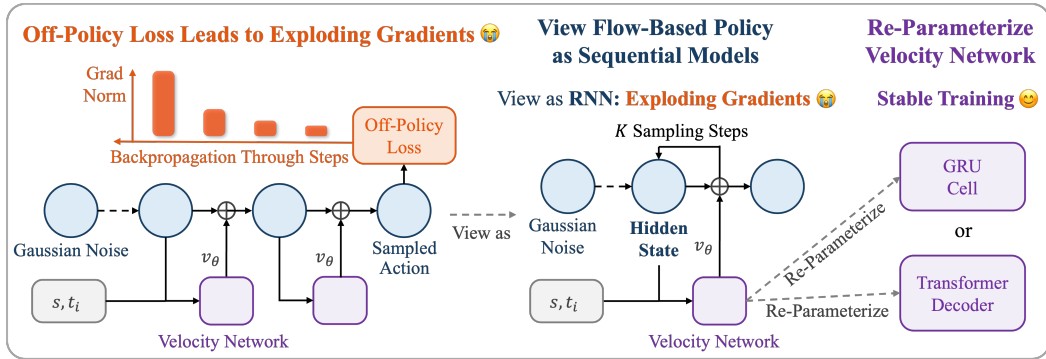

Figure 1: An Overview of SAC Flow. The multi-step sampling process of flow-based policies frequently causes exploding gradients during off-policy RL updates. Our key insight is to treat the flow-based policy as a sequential model, for which we first demonstrate an algebraic equivalence to an RNN. We then reparameterize the flow's velocity network using modern sequential architectures (e.g., GRU, Transformer). Our approach stabilizes off-policy RL training and achieves state-of-the-art performance.

## ABSTRACT

Training expressive flow-based policies with off-policy reinforcement learning is notoriously unstable due to gradient pathologies in the multi-step action sampling process. We trace this instability to a fundamental connection: the flow rollout is algebraically equivalent to a residual recurrent computation, making it susceptible to the same vanishing and exploding gradients as RNNs. To address this, we reparameterize the velocity network using principles from modern sequential models, introducing two stable architectures: Flow-G, which incorporates a gated velocity, and Flow-T, which utilizes a decoded velocity. We then develop a practical SAC-based algorithm, enabled by a noise-augmented rollout, that facilitates direct end-to-end training of these policies. Our approach supports both from-scratch and offline-to-online learning and achieves state-of-the-art performance on continuous control and robotic manipulation benchmarks, eliminating the need for common workarounds like policy distillation or surrogate objectives. Anonymized code is available at https://anonymous.4open.science/r/SAC-FLOW.

.

# 1 INTRODUCTION

Flow-based policies have shown strong potential on challenging continuous-control tasks, including robot manipulation, due to their ability to represent rich, multimodal action distributions (Black et al., 2024; Lipman et al., 2022; Jiang et al., 2025). Early successes predominantly arose in imitation learning, where a flow-based policy is trained to reproduce expert behavior from static datasets (Luo et al., 2025; Tarasov et al., 2025). However, pure behavior cloning is fundamentally limited: dataset coverage is often sparse and of mixed quality (Kim et al., 2024; Garcia et al., 2025), and the lack of environment interaction prevents exploration, making it difficult to exceed demonstrator performance on hard tasks (Belkhale et al., 2023; Zare et al., 2024).

A natural next step is to train flow-based policies with reinforcement learning. On-policy variants of PPO adapted to flows have demonstrated strong returns, yet they remain sample-inefficient (Schulman et al., 2017; Zhang et al., 2025). Off-policy methods promise much higher data efficiency and early integrations with flow-based policies on MuJoCo and DeepMind Control show encouraging results (Todorov et al., 2012; Tunyasuvunakool et al., 2020; Lv et al., 2025; Park et al., 2025). However, these successes typically come with design compromises that leave a central issue unresolved. Either the update relies on surrogate objectives that avoid differentiating through the rollout of the original flow, or the flow is distilled into a simpler one-step actor that can be optimized with standard off-policy losses. Both strategies reduce gradient stress but decouple optimization from the expressive generator and tend to blunt the benefits of multimodal flow-based policies (Park et al., 2025; Lv et al., 2025).

We propose a different viewpoint: treat the flow-based policy as a sequential model. Concretely, we show that the Euler integration used to generate actions in the flow-based policy is algebraically identical to the recurrent computation of a residual RNN. This observation explains the instability observed with off-policy training: the same vanishing or exploding gradients known to affect RNNs also afflict the flow rollout. Building on this link, we reparameterize the vanilla velocity network with the cell of modern sequential models that are designed to stabilize deep recurrent computations. We introduce two such novel designs of the flow-based policy: Flow-G, which incorporates a GRU-style gated velocity to regulate gradient flow across rollout steps, and Flow-T, which utilizes a Transformer-style decoded velocity to refine the action-time token via state-only cross-attention and a residual feed-forward network.

Our main contributions are summarized as follows:

- **A sequential model perspective for stable flow-based policies.** We formalize the $K$-step flow rollout as a residual RNN computation, providing a clear theoretical explanation for the gradient pathologies that cause instability in off-policy training. This insight allows us to reparameterize the velocity network with modern sequential architectures, leading to two novel, stable designs: **Flow-G** (GRU-gated) and **Flow-T** (Transformer-decoded). Our approach resolves critical gradient pathologies, enabling direct end-to-end optimization and eliminating the need for surrogate objectives or policy distillation.

- **A practical and sample-efficient SAC framework for flow policies.** We develop SAC Flow, a robust off-policy algorithm built upon our stabilized architectures. By introducing a noise-augmented rollout, we enable tractable likelihood computation for the SAC objective, a key technical hurdle. This approach yields two robust training procedures: (i) a stable from-scratch trainer for dense-reward tasks and (ii) a unified offline-to-online pipeline for sparse-reward tasks.

- **Extensive experimental evaluation.** We demonstrate the effectiveness of SAC Flow across multiple benchmarks. In from-scratch training on challenging MuJoCo tasks, our approach delivers performance gains of up to $130\%$ over strong baselines. Furthermore, in complex offline-to-online manipulation tasks on OGBench, it achieves up to a $60\%$ higher success rate. These results empirically validate the superior **sample efficiency** of our direct off-policy training approach, with ablation studies further confirming the robustness of our designs.

## 2 PRELIMINARIES

### 2.1 REINFORCEMENT LEARNING

We consider policy optimization in an infinite-horizon Markov decision process $\langle \mathcal{S}, \mathcal{A}, p, r, \rho \rangle$ with continuous state and action spaces. The transition function $p : \mathcal{S} \times \mathcal{A} \times \mathcal{S} \to [0, \infty)$ specifies the transition probability density, and rewards are $r_h = r(s_h, a_h) \in [r_{\min}, r_{\max}]$, where $a_h$ is sampled from the policy $\pi(\cdot|s_h)$. The objective of reinforcement learning is to learn an optimal policy $\pi^*$ that maximizes the expected cumulative reward, $\pi^* = \arg\max_\pi \mathbb{E}^\pi \left[ \sum_{h=0}^\infty \gamma^h r_h \right]$.

### 2.2 SOFT ACTOR-CRITIC ALGORITHM

To encourage policies to maintain stochasticity and explore more effectively, the standard objective is augmented with an entropy term, $\hat{J}(\pi) = \mathbb{E}^\pi [\sum_{h=0}^\infty \gamma^h (r_h + \alpha \mathcal{H})]$, where $\mathcal{H}(\pi(\cdot \mid s_h)) = -\mathbb{E}_{a \sim \pi(\cdot|s_h)} [\log \pi(a \mid s)]$ denotes the state-conditional policy entropy. In this setting, the Soft Actor-Critic algorithm (Haarnoja et al., 2018) is introduced to optimize this objective. The target $\hat{J}(\pi)$ is typically approximated with the soft $Q$-function $Q_\psi(s_h, a_h)$, which is updated through the TD loss:

$$L(\psi) = \left[ Q_\psi\left(s_h, a_h\right) - \left(r_h + \gamma Q_{\bar{\psi}}\left(s_{h+1}, a_{h+1}\right) - \alpha \log \pi_\theta\left(a_{h+1} \mid s_{h+1}\right)\right) \right]^2, \tag{1}$$

where $a_{h+1} \sim \pi_\theta\left(\cdot \mid s_{h+1}\right)$, $(s_h, a_h, s_{h+1}, r_h)$ are sampled from the replay buffer, and $\bar{\psi}$ is a delayed copy of $\psi$ through which gradients do not flow for stability. To maximize the soft $Q$-function $Q_\psi(s_h, a_h)$, the policy $\pi_\theta$ is updated through

$$L(\theta) = \alpha \log \pi_\theta\left(a_h^\theta \mid s_h\right) - Q_\psi\left(s_h, a_h^\theta\right), \quad a_h^\theta \sim \pi_\theta\left(\cdot \mid s_h\right). \tag{2}$$

Here, $a_h^\theta$ highlights a reparameterized action sample that allows gradients to propagate from the policy to the action, in contrast to the TD update, where the action is detached.

### 2.3 FLOW-BASED POLICY IN REINFORCEMENT LEARNING

Gaussian policies are the standard choice in continuous-control RL (Yang et al., 2021; Ziesche & Rozo, 2024), yet a single unimodal Gaussian cannot capture inherently multimodal action distributions. This limitation is especially harmful in long-horizon robotic control, as such tasks often benefit from policies that output temporally extended actions (i.e., "action chunking") (Li et al., 2025). The distribution over these action sequences is often inherently multimodal, which a unimodal Gaussian fails to represent. Diffusion policies alleviate this by modeling arbitrary normalizable distributions and have achieved state-of-the-art results on manipulation benchmarks (Bekris et al., 2025; Wang et al., 2022; Ren et al., 2024), but their iterative denoising makes both training and inference expensive. Recently, flow-based policies have emerged as a simpler alternative: trained with flow-matching objectives, they offer easier training and faster inference while often matching or exceeding diffusion quality (Lipman et al., 2022; Park et al., 2025; Zhang et al., 2025).

A flow-based policy transports a simple, state-conditioned base $p_0(\cdot \mid s)$ over the action space $\mathcal{A} = \mathbb{R}^d$ to a target policy $p_1(\cdot \mid s)$ via a time-indexed map $\varrho : [0, 1] \times \mathcal{A} \times \mathcal{S} \to \mathcal{A}$, with $A_t := \varrho_t(A_0 \mid s)$ for $t \in [0, 1]$, where $A_0 \sim p_0(\cdot \mid s)$ and $A_1 \sim p_1(\cdot \mid s)$. The trajectory satisfies the ODE $\frac{d}{dt} \varrho_t(A_0 \mid s) = v(t, \varrho_t(A_0 \mid s), s)$, where $v$ is a learnable velocity field. We adopt Rectified Flow (Liu et al., 2022), which uses the straight path $A_t = (1 - t)A_0 + tA_1$ and the standard Gaussian base $p_0(\cdot \mid s) = \mathcal{N}(0, I_d)$. In this case $v(t, A_t, s) = \frac{d}{dt} A_t = A_1 - A_0$, yielding the flow-matching objective

$$\hat{\theta} = \arg\min_\theta \mathbb{E}_{\substack{A_0 \sim \mathcal{N}(0, I_d), (A_1 = a, s) \sim \mathcal{D}, \\ t \sim \mathrm{Unif}[0,1]}} \left[ \left\| A_1 - A_0 - v_\theta(t, (1 - t)A_0 + tA_1, s) \right\|_2^2 \right], \tag{3}$$

where $\mathcal{D}$ denotes the dataset of state–action pairs. In inference, the learned field is integrated numerically with flow rollout to obtain:

$$A_{t_{i+1}} = A_{t_i} + \Delta t_i \, v_\theta(t_i, A_{t_i}, s), \quad 0 = t_0 < \cdots < t_K = 1, \tag{4}$$

where $\Delta t_i = t_{i+1} - t_i$. The resulting distribution over $A_1$ induced by $A_0 \sim \mathcal{N}(0, I_d)$ is denoted $\mu_\theta(\cdot \mid s)$ and serves as the stochastic policy $a = A_1 \sim \pi_\theta(\cdot \mid s)$.

Flow-based policies can be trained offline from demonstrations using Equ. (3), and they can also be optimized with RL. **On-policy** methods (e.g., PPO-style training tailored to flows (Zhang et al., 2025; Ren et al., 2024; Psenka et al., 2024)) attain strong performance on challenging robotics tasks but remain sample-inefficient. **Off-policy** methods (e.g., SAC, TD3) are highly sample-efficient (Mambelli et al., 2024), yet directly backpropagating through the $K$-step action sampling is often unstable, especially for large $K$ (Park et al., 2025). To mitigate this, prior work either distills a flow-based policy into a simpler actor trained with standard off-policy losses (Park et al., 2025) or proposes surrogate off-policy objectives that train the velocity field without differentiating through the full flow rollout (Lv et al., 2025).

We take a different route. We recast the flow rollout as a sequential model and redesign the velocity parameterization accordingly. We introduce Flow-G, which uses a GRU-style gated velocity, and Flow-T, which uses a Transformer-style decoded velocity. These parameterizations stabilize the $K$-step backpropagating and allow direct off-policy training of the flow-based policy. We instantiate the framework with SAC, and the same formulation applies to other off-policy algorithms.

## 3   FROM FLOW ROLLOUT TO SEQUENTIAL MODELS

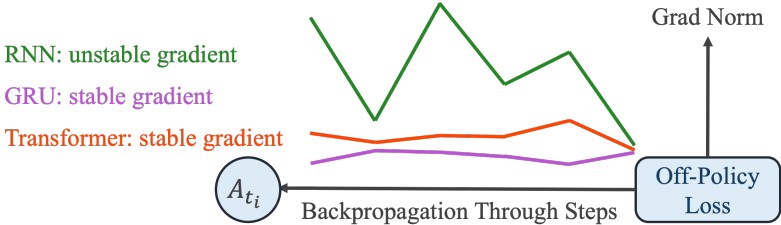

Figure 2: An illustration of gradient norms during training. By conceptualizing a flow-based model as an RNN, the most basic sequential models, we observe that it still suffers from the exploding gradients during training. This motivates our work to model the flow-based model as advanced sequential architectures, such as a GRU or a Transformer. These models can be updated with stable gradients during the backpropagation process.

In this section, we reveal a key insight: flow-based policies are fundamentally sequential models. As conceptually illustrated in Fig. 2, standard flow rollouts exhibit gradient instabilities analogous to vanilla RNNs, while modern sequential architectures offer more stable gradient flow, motivating our velocity network designs.

**Flow-based policy as RNN (Fig. 3a).** Treat the intermediate action $A_{t_i}$ as the hidden state and $(t_i, s)$ as the input. Then Equ. (4) is a residual RNN step (Goel et al., 2017):

$$A_{t_{i+1}} = A_{t_i} + f_\theta(t_i, A_{t_i}, s), \quad \text{with } f_\theta(\cdot) = \Delta t_i \, v_\theta(\cdot), \tag{5}$$

where $f_\theta(\cdot)$ denotes the RNN cell. Consequently, training a flow-based policy with off-policy losses backpropagates through a deep recurrent stack of $K$ updates in RNN, which is prone to gradient explosion and vanishing (Bengio et al., 1994; Pascanu et al., 2013). This explains the instability observed when naively applying off-policy reinforcement learning to standard flow-based policies.

**Flow-based policy as GRU (Flow-G, Fig. 3b).** To improve gradient stability, we endow the velocity with a GRU-style update gate. Let $g_i = Sig(z_\theta(t_i, A_{t_i}, s))$ and let $\hat{v}_\theta$ be a candidate network. Define

$$A_{t_{i+1}} = A_{t_i} + \Delta t_i \left( g_i \odot (\hat{v}_\theta(t_i, A_{t_i}, s) - A_{t_i}) \right), \tag{6}$$

where $\odot$ denotes elementwise multiplication and $Sig(\cdot)$ is the logistic sigmoid. Equ. (6) is exactly a flow sampling step with gated velocity $v_\theta = g_i \odot (\hat{v}_\theta - A_{t_i})$, which mirrors the structure of the update in a GRU cell but expressed in the velocity parameterization used by the flow rollout. The gate network $g_i$ adaptively interpolates between keeping the current intermediate action and forming a new one.

**Flow-based policy as Transformer (Flow-T, Fig. 3c).** We parameterize the velocity function $v_\theta$ using a Transformer architecture conditioned on the environment state $s$. To maintain the Markov property of the flow, we depart from a traditional causal, autoregressive formulation. Instead, the model first computes independent embeddings for the current action-time token $A_{t_i}$ and a single, global embedding for the state $s$:

$$\Phi_{A_i} = E_A\big(\phi_t(t_i), A_{t_i}\big), \qquad \Phi_S = E_S\big(\phi_s(s)\big), \tag{7}$$

where $E_A$ and $E_S$ are linear projections. Within the Decoder layers, a diagonal mask is applied to the self-attention mechanism, effectively reducing it to a position-wise transformation that processes each token $\Phi_{A_i}$ independently, without mixing information across the time steps $i$. The crucial step for context integration is a dedicated cross-attention module, where each action token $\Phi_{A_i}$ queries the shared state embedding $\Phi_S$. A stack of $L$ pre-norm residual blocks refines the action tokens:

$$Y_i^{(l)} = \Phi_{A_i}^{(l-1)} + \text{Cross}_l\Big(\text{LN}(\Phi_{A_i}^{(l-1)}), \text{context} = \text{LN}(\Phi_S)\Big), \ \Phi_{A_i}^{(l)} = Y_i^{(l)} + \text{FFN}_l\big(\text{LN}(Y_i^{(l)})\big), \tag{8}$$

for layers $l = 1, \ldots, L$, where $\Phi_{A_i}^{(0)} := \Phi_{A_i}$. Each block is completed by a feed-forward network, and the final representation is projected to the velocity space:

$$A_{t_{i+1}} \;=\; A_{t_i} \;+\; \Delta t_i\, W_o\Big(\text{LN}(\Phi_{A_i}^{(L)})\Big), \tag{9}$$

where $W_o$ is a linear projection and $v_\theta(t_i, A_{t_i}, s) = W_o\Big(\text{LN}(\Phi_{A_i}^{(L)})\Big)$ is the decoded velocity in Flow-T. Each velocity evaluation therefore executes $L$ layers that refine the current action token $\Phi_{A_i}$ based on the global state context from $\Phi_S$, not on a causal history of other tokens. This state-conditioned refinement of the entire trajectory maintains the fundamental Markov property of flow-based policy while enabling stable integration with off-policy learning algorithms.

**Takeaway for off-policy reinforcement learning.** Equ. (5) establishes that a standard flow rollout is a residual recurrent computation. Introducing a gate network leads to Flow-G in Equ. (6), which improves gradient stability. Replacing the velocity with the normalized residual block in Equ. (9) yields Flow-T. This architecture provides well-conditioned depth and, crucially, aggregates context with the well-established Transformer architectures.

The core technical motivation is to stabilize the recurrent computation in Equ. (5), which suffers from the exploding/vanishing gradient problem due to unstable Jacobian products during backpropagation. Our Flow-G and Flow-T designs directly mitigate this via stabilizing mechanisms. A detailed mathematical analysis of this instability and our solution is provided in Appendix B.

These parameterizations serve as drop-in replacements for $v_\theta$ in Equ. (4) without altering the surrounding algorithm. As a result, they enable direct and stable off-policy training with methods such as SAC, remove the need for auxiliary distillation actors and surrogate objectives, and keep flow rollout efficient at test time.

## 4 Training Flow-based Policy via SAC

With gradient stability achieved through our sequential parameterizations (Flow-G and Flow-T), we can now train flow-based policies directly with off-policy reinforcement learning. The key technical challenge is computing policy likelihoods for the K-step rollout in Equ. (4)—a requirement for the entropy-regularized objective in SAC. We solve this through a principled noise-augmented rollout that preserves the final action distribution while enabling tractable per-step likelihood computation.

**Likelihood via a noise-augmented rollout.** SAC requires explicit policy likelihoods for entropy regularization, but the deterministic $K$-step rollout in Equ. (4) yields intractable densities. We address this by making the rollout stochastic while preserving the marginal of the final action, which induces a product of per-step Gaussian transitions and a tractable joint path density $p_c(\mathcal{A} \mid s)$ over intermediate actions $\mathcal{A} = (A_{t_0}, \ldots, A_{t_K})$. The construction details are deferred to Appendix A; here we use the resulting $\log p_c(\mathcal{A} \mid s)$ as a drop-in entropy term.

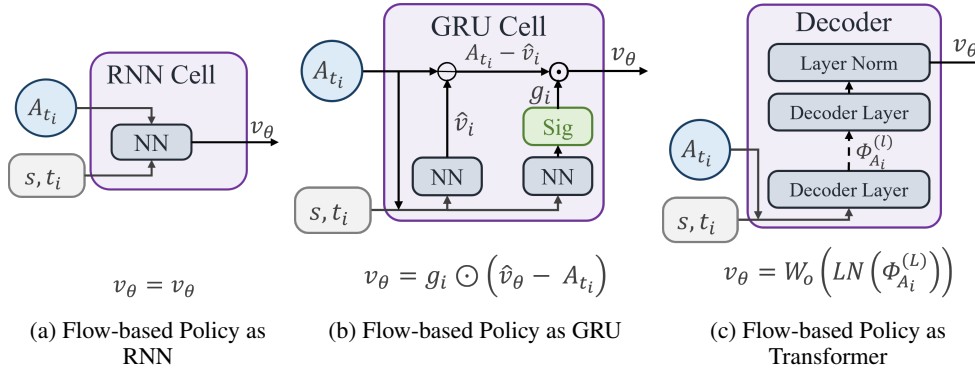

$$v_\theta = v_\theta$$

(a) Flow-based Policy as RNN

$$v_\theta = g_i \odot \left( \hat{v}_\theta - A_{t_i} \right)$$

(b) Flow-based Policy as GRU

$$v_\theta = W_o \left( LN \left( \Phi_{A_i}^{(L)} \right) \right)$$

(c) Flow-based Policy as Transformer

Figure 3: Velocity network parameterizations for the flow-based policy, shown in the view of sequential models. (a) **RNN Cell**: It represents the standard flow-based policy where the velocity $v_\theta$ is the direct output of a neural network. This simple formulation is prone to gradient instability. (b) **GRU Cell**: The velocity is computed using a GRU-style gated mechanism. A gate $g_i$ adaptively controls the update strength from a candidate network $\hat{v}_i$, which stabilizes gradient flow. (c) **Decoder**: The velocity is modeled using a Transformer decoder, where the action-time token $A_{t_i}$ is refined through $L$ layers of state-conditioned cross-attention to produce a decoded velocity.

**From-scratch training.** With tractable likelihoods established, the SAC losses become straightforward. Given a critic $Q_\psi$ and a flow-based policy $\pi_\theta$ (with Flow-G or Flow-T as $v_\theta$), we optimize:

$$L_{\text{actor}}(\theta) = \alpha \log p_c(\mathcal{A}^\theta \mid s_h) - Q_\psi(s_h, a_h^\theta), \quad \mathcal{A}^\theta \sim \pi_\theta(\cdot \mid s_h), \quad a_h^\theta = \tanh(A_{t_K}^\theta), \qquad (10)$$

$$L_{\text{critic}}(\psi) = \left[ Q_\psi(s_h, a_h) - \left( r_h + \gamma Q_{\bar\psi}(s_{h+1}, a_{h+1}) - \alpha \log p_c(\mathcal{A}_{h+1} \mid s_{h+1}) \right) \right]^2, \qquad (11)$$

where $(s_h, a_h, r_h, s_{h+1})$ comes from the replay buffer, $\mathcal{A}_{h+1}, a_{h+1} \sim \pi_\theta(\cdot \mid s_{h+1})$, and $\bar\psi$ is a delayed copy.

**Offline-to-online training.** For sparse-reward tasks where expert demonstrations are available, we modify the actor loss to include a proximity regularizer:

$$L_{\text{actor}}^o(\theta) = \alpha \log p_c(\mathcal{A}^\theta \mid s_h) - Q_\psi(s_h, a_h^\theta) + \beta \| a_h^\theta - a_h \|_2^2, \quad (s_h, a_h) \sim \mathcal{B}. \qquad (12)$$

This approach begins with flow-matching pretraining on expert data via Equ. (3), then transitions to online learning while maintaining proximity to the replay buffer. The complete procedures are summarized in Algos. 1 and 2.

---

**Algorithm 1** SAC Flow (from scratch)

---

1: Initialize critic $Q_\psi$, target $Q_{\bar\psi}$, flow-based policy $\pi_\theta$ with Flow-G or Flow-T; replay buffer $\mathcal{B}$.
2: **for** each update **do**
3:     Interact with the environment using $\pi_\theta$; push $(s_t, a_t, r_t, s_{t+1})$ to $\mathcal{B}$.
4:     Sample $\{(s_h, a_h, r_h, s_{h+1})\}_{h=1}^N \sim \mathcal{B}$.
5:     Actor: draw $a_h^\theta$ by a $K$-step noisy rollout; minimize Equ. (10).
6:     Critic: minimize Equ. (11); update target by an exponential moving average.
7: **end for**

---

For clarity, we refer to our methods as SAC Flow-G and SAC Flow-T, corresponding to training with Flow-G and Flow-T via SAC, respectively. Both terms apply to both from-scratch and offline-to-online training variants.

## 5 EXPERIMENT

We conduct extensive experiments on locomotion and manipulation benchmarks to validate our approach. The evaluation encompasses: (1) experimental setup and baseline comparisons for from-scratch and offline-to-online training, (2) performance benchmarking of SAC Flow-G and SAC

---

**Algorithm 2** SAC Flow (offline-to-online)

1: Initialize $Q_\psi$, $Q_{\bar\psi}$, $\pi_\theta$; set $\mathcal{B} \leftarrow \mathcal{D}_{\text{expert}}$.
2: **for** $\ell = 1$ to $L_{\text{off}} + L_{\text{on}}$ **do**
3:    **if** $\ell > L_{\text{off}}$ **then**
4:       Interact with the environment using $\pi_\theta$; append to $\mathcal{B}$.
5:    **end if**
6:    Sample $\{(s_h, a_h, r_h, s_{h+1})\}_{h=1}^N \sim \mathcal{B}$.
7:    Actor: minimize Equ. (12) with $a_h^\theta$ from the noisy rollout.
8:    Critic: minimize Equ. (11); update the target network.
9:    **if** $\ell \le L_{\text{off}}$ **then**
10:       Flow-matching pretraining via Equ. (3).
11:    **end if**
12: **end for**

---

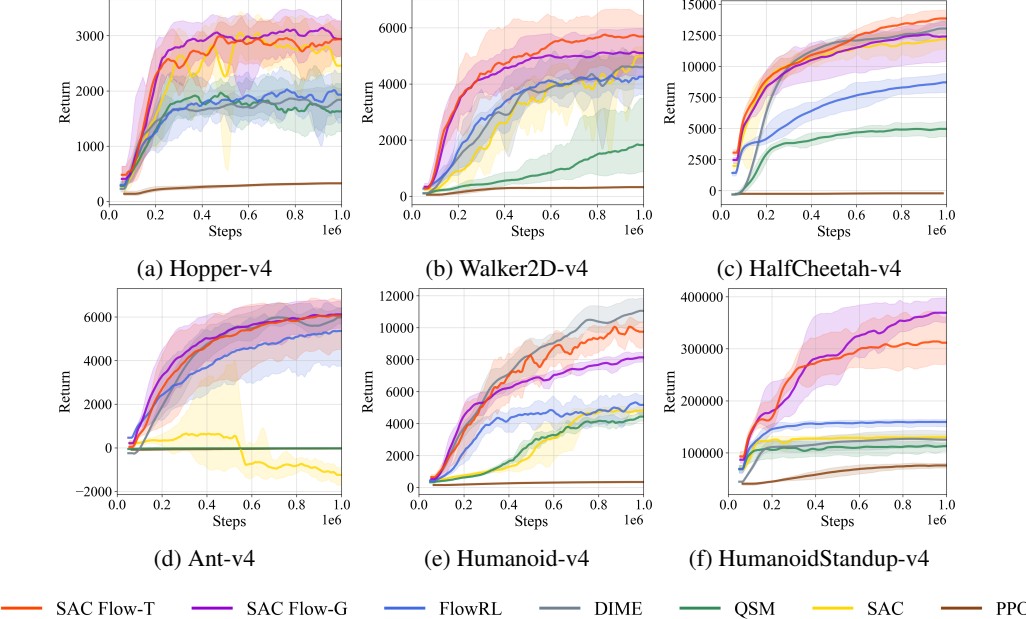

(a) Hopper-v4          (b) Walker2D-v4       (c) HalfCheetah-v4

(d) Ant-v4            (e) Humanoid-v4       (f) HumanoidStandup-v4

SAC Flow-T    SAC Flow-G    FlowRL    DIME    QSM    SAC    PPO

Figure 4: From-scratch training performance. Our SAC Flow-T and SAC Flow-G achieve comparable or better performance accross all tasks except Humanoid (Fig. (a)-(f)), demonstrating significant sample efficiency and convergence stability.

Flow-T against recent methods, and (3) ablation studies analyzing the effectiveness of our design components. All results are averaged over 5 random seeds and use the 95% confidence interval.

## 5.1 SETTINGS

### 5.1.1 ENVIRONMENTS AND OFFLINE DATASETS

We evaluate our method on three benchmarks for locomotion and robotic manipulation: **MuJoCo** (Todorov et al., 2012; Brockman et al., 2016), **OGBench** (Park et al., 2024), and **Robomimic** (Mandlekar et al., 2021). MuJoCo tasks, which feature dense rewards, are used to evaluate from-scratch learning performance. Then we conduct offline-to-online experiments on OGBench and Robomimic, using their respective official offline datasets[1].

---

[1]OGBench: `https://github.com/seohongpark/ogbench`, Robomimic: `https://robomimic.github.io/docs/datasets/robomimic_v0.1.html`

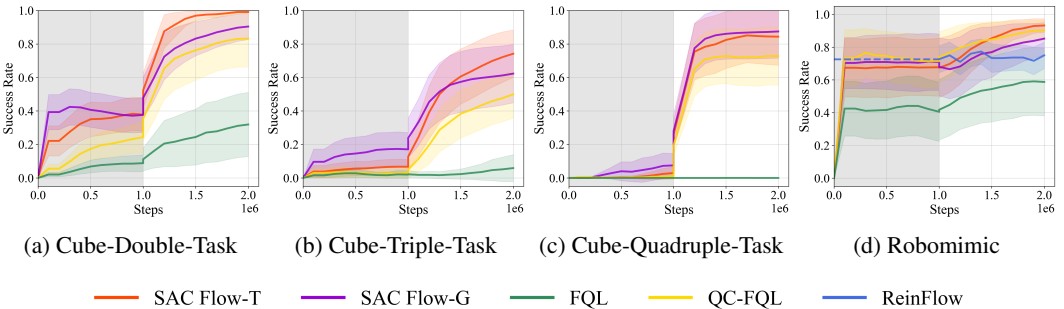

Figure 5: Aggregated offline-to-online performance on OGBench and Robomimic benchmarks. Each curve shows the mean success rate averaged across multiple task instances within a domain. Specifically, the **OGBench** results for *Cube-Double*, *Triple*, and *Quadruple* (a-c) are each aggregated over five distinct single-task environments. The **Robomimic** result (d) is aggregated across the *Lift*, *Can*, and *Square* tasks.

### 5.1.2  Baselines

For the from-scratch training, we compare SAC-Flow against five baselines. **(1) Q-score matching (QSM)** (Psenka et al., 2024) directly optimizes the diffusion policy's score function using the gradient of the Q-function. **(2) DIME** (Celik et al., 2025) is a representative max-entropy RL method for diffusion policy, addressing the challenge of entropy calculation. **(3) FlowRL** (Lv et al., 2025) is the state-of-the-art (SOTA) method, which trains a flow-based policy by directly maximizing the Q-value, regularized by a Wasserstein-2 constraint. Finally we apply two classical RL algorithms: **(4) SAC** (Haarnoja et al., 2018) and **(5) PPO** (Schulman et al., 2017), with Gaussian policies as fundamental from-scratch baselines.

To evaluate the offline-to-online capability, we select three baselines, including on-policy and off-policy methods. **(1) ReinFlow** (Zhang et al., 2025) solves the difficulty of calculating log probability through multi-step flow inference, enabling on-policy PPO update for flow-based policy. It should be noted that ReinFlow is only tested in Robomimic due to a lack of official implementation for its use in OGBench. **(2) Flow Q-Learning (FQL)** (Park et al., 2025) uses SAC-style update to achieve high data-efficient RL tuning. FQL uses a one-step policy to estimate the flow model, avoiding the instability of backpropagation through time. And its successor, **(3) Q-chunking FQL (QC-FQL)** (Li et al., 2025), extends FQL to handle action chunking by operating in temporally extended action spaces.

Among all experiments, the sampling steps of flow-based policies are set to 4, and the denoising steps of diffusion policies are set to 16. More details of the experimental setting are described in Appendix D and Appendix E.

### 5.2  Main results

Fig. 4 illustrates the results for from-scratch training. Our methods, SAC Flow-G and SAC Flow-T, achieve superior or comparable performance across most MuJoCo tasks, with the exception of Humanoid. Although DIME and FlowRL generally converge faster than other baselines, our methods consistently surpass FlowRL, benefiting from direct optimization of the SAC objective. Furthermore, SAC Flow outperforms DIME in Hopper (Fig. 15a), Walker (Fig. 15b), and HumanoidStandup (Fig. 4f), while achieving comparable results in HalfCheetah (Fig. 15c) and Ant (Fig. 15d). Moreover, with the expressive parameterization of flow-based policy, our method achieves much higher final performance in challenging tasks, demonstrating up to a $130\%$ improvement over the baseline (Fig. 4f), and remains convergence stability in simple tasks (Fig. 15a, 15b, and 15c). For reference, we include the on-policy baseline, PPO, to highlight the superior sample efficiency of off-policy algorithms. Finally, we find that all from-scratch methods struggle in tasks with large exploration spaces and sparse rewards, such as Robomimic-Can and OGBench-cube (see Appendix F.1, Fig. 14), underscoring the necessity of an offline-to-online training setting.

Fig. 5 shows the offline-to-online training performance in sparse reward tasks. All methods are trained on 1M offline updates followed by 1M online steps. In the challenging OGBench environments, including cube-triple and cube-quadruple, our proposed methods, particularly SAC Flow-T, achieve rapid convergence and attain a state-of-the-art overall success rate. In the Robomimic environment, however, SAC Flow-T and SAC Flow-G only yield results comparable to QC-FQL. This is primarily because the training is strictly regularized with a large $\beta$ value (Equ. (12)). As a result, the learning capacity of the flow model is severely limited, causing its performance to be similar to that of the one-step policy in QC-FQL. We further compare the on-policy baseline, Reinflow, in Robomimic. Leveraging the high data efficiency of off-policy learning, our SAC Flow-G and SAC Flow-T outperform Reinflow under 1M online steps. The additional results are available in Appendix F.

## 5.3 ABLATION STUDY

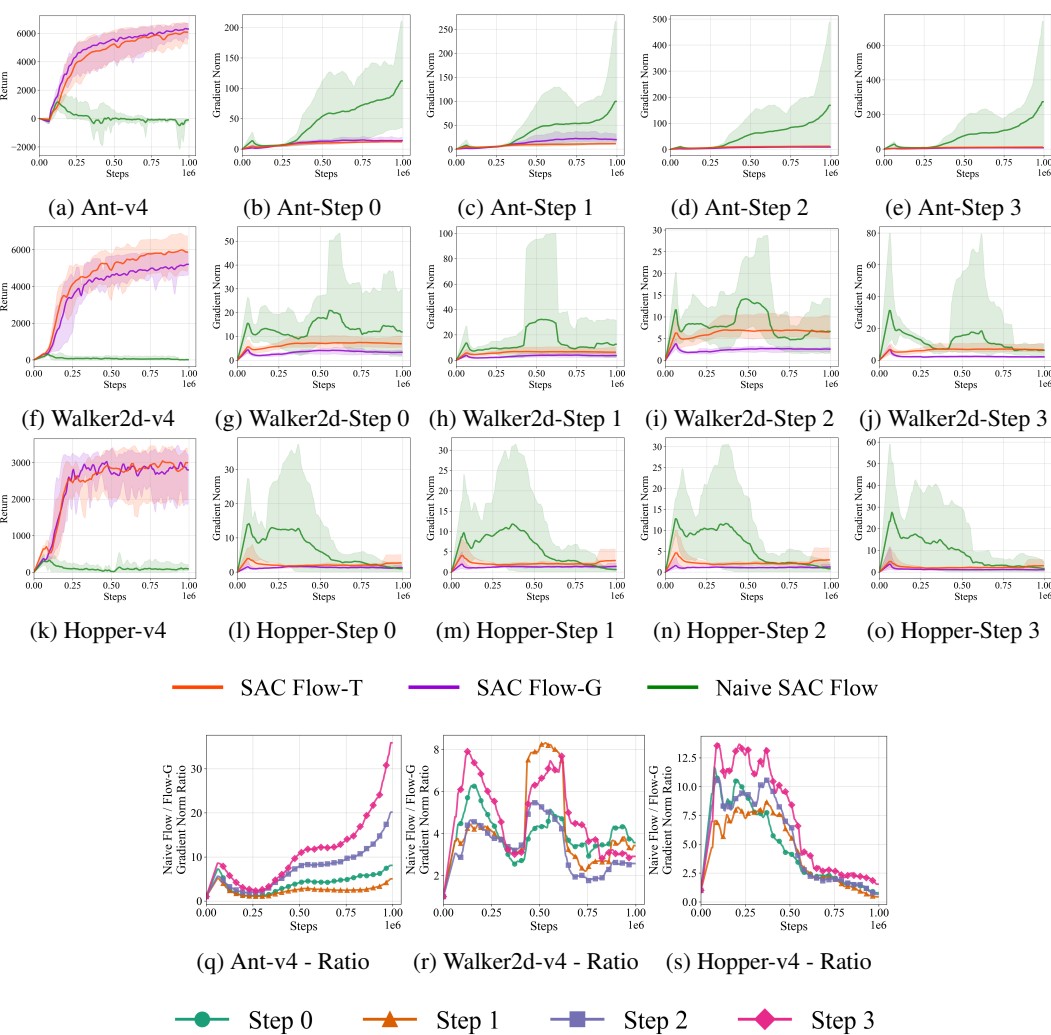

Figure 6: Ablation study on velocity network parameterizations. Our SAC Flow-T and SAC Flow-G significantly reduce the gradient exploding and enable stable training. The first column (a, f, k) displays the episodic return for the environments. The subsequent four columns illustrate the gradient norms for steps 0 through 3 of the flow sampling process, respectively. The bottom row (q–s) visualizes the gradient norm ratio (Naive / Flow-G), revealing that across all tasks, the Naive gradient norm explodes to approximately tenfold the stable magnitude around the time of performance collapse ($10^5$ steps).

**Ablation study on velocity network parameterizations.** We begin by analyzing the gradient dynamics of our proposed architectures, SAC Flow-G and SAC Flow-T. We benchmark these against a Naive SAC Flow baseline that utilizes a standard MLP velocity parameterization without sequential modeling. As illustrated in the first three rows of Fig. 6, the naive baseline exhibits severe gradient pathologies, characterized by erratic norm oscillations along the backpropagation path (specifically from sampling step $k = 3$ back to $k = 0$). In contrast, our methods maintain well-conditioned gradient norms across the entire rollout. This instability in the Naive SAC Flow directly precipitates performance degradation, as evidenced by its failure to learn in the Ant, Walker2d, and Hopper. (Figs. 6a, 6f, and 6k).

To provide a unified explanation despite the varying absolute gradient scales across tasks (e.g., gradients in Ant are naturally larger than in Walker2d), we further analyze the relative stability by computing the gradient norm ratio (Naive SAC Flow / SAC Flow-G), shown in the bottom row of Fig. 6 (q–s). This metric reveals a striking consistency: across all distinct tasks, the gradient norm of the Naive baseline escalates to approximately tenfold that of the stable SAC Flow-G. Crucially, this 10-fold relative explosion typically peaks around $10^5$ steps, which aligns perfectly with the inflection point where the Naive baseline's performance stagnates and begins to deteriorate. These empirical results conclusively validate that the standard flow rollout suffers from severe relative gradient instability, and our sequential reparameterizations effectively mitigate this issue to enable stable training.

**Ablation study on flow sampling steps.** Fig. 7 shows the performance of SAC FLow-T and SAC Flow-G under sampling steps $K = 4, 7, 10$. A larger number of sampling steps can further challenge the stability of gradient backpropagation. The experiments show that our approach, especially SAC Flow-T, is robust to the number of sampling steps.

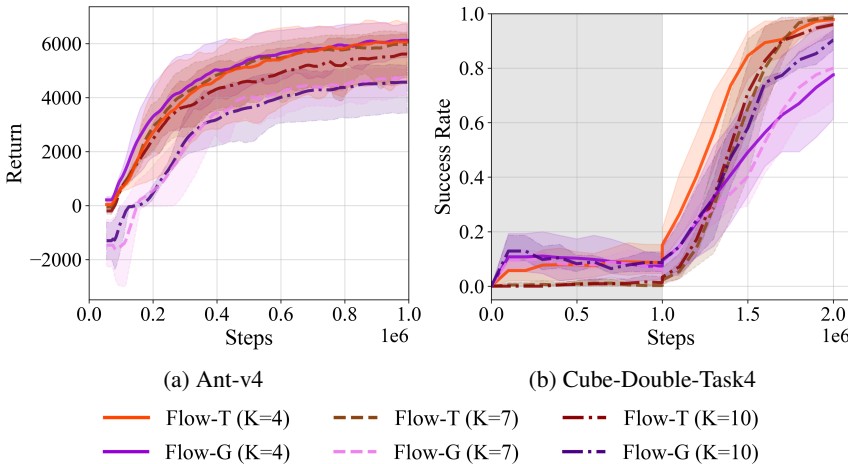

(a) Ant-v4  (b) Cube-Double-Task4

Figure 7: Ablation study on flow sampling steps. Our SAC Flow-G and SAC Flow-T are robust to the number of sampling steps.

## 6 CONCLUSION

In this paper, we introduce SAC Flow, a sample-efficient and high-performance off-policy RL algorithm for flow-based policies. SAC Flow addresses the issue of gradient instability in training flow-based policies by treating the flow-based model as a sequential model and reparameterizing its velocity network as a GRU or a Transformer. We evaluate the performance of SAC Flow in both from-scratch and offline-to-online training settings. SAC Flow demonstrates rapid convergence and achieves state-of-the-art performance across multiple locomotion and manipulation tasks.

Looking forward, we will validate SAC Flow on real robots and explore lighter sequential parameterizations with structure-aware updates, while studying sim-to-real robustness, tighter stability guarantees, and risk-aware objectives for reliable deployment.

## ETHICS STATEMENT

We adhere to the ICLR Code of Ethics. Our research introduces foundational RL algorithms evaluated exclusively on public benchmarks and datasets. This work does not involve human subjects or sensitive data and presents no foreseeable direct ethical concerns.

## REPRODUCIBILITY STATEMENT

To ensure reproducibility, additional implementation details, hyperparameters, and experimental results are provided in the Appendix. All experiments leverage public benchmarks and datasets. The anonymous code is available at `https://anonymous.4open.science/r/SAC-FLOW`

### ACKNOWLEDGMENTS

We would like to thank Yuze Sheng for assistance with the real-robot experiments of SAC Flow. This work was supported by the National Natural Science Foundation of China (No. 62406159, 62325405), Shenzhen Science and Technology Program (No. KJZD20240903100905008), Zhongguancun Academy (Grant No. C20250301), Beijing National Research Center for Information Science and Technology (BNRist), Beijing Innovation Center for Future Chips, and Ant Group.

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

# A  THE DERIVATION OF SAC LOSS IN THE FLOW-BASED POLICY

This appendix consolidates and expands our derivations for training SAC on a $K$-step flow roll-out, including the likelihood construction via a noise-augmented rollout, the joint path density, the pathwise score expansion, gradients for actor/critic, the temperature update, and practical notes for implementation.

## A.1  NOISE-AUGMENTED ROLLOUT AND DRIFT CORRECTION

We start from the deterministic $K$-step Euler rollout in Equ. (4):

$$A_{t_{i+1}} = A_{t_i} + \Delta t_i\, v_\theta(t_i, A_{t_i}, s), \quad 0 = t_0 < \cdots < t_K = 1.$$

For likelihood-based training, we convert it into a stochastic rollout that leaves the final marginal invariant by adding isotropic Gaussian noise with a compensating drift (Holderrieth & Erives, 2025):

$$A_{t_{i+1}} = A_{t_i} + b_\theta(t_i, A_{t_i}, s)\, \Delta t_i + \sigma_\theta \sqrt{\Delta t_i}\, \varepsilon_i, \quad \varepsilon_i \sim \mathcal{N}(0, I_d). \tag{13}$$

A convenient drift that matches rectified-flow families is

$$b_\theta(t_i, A_{t_i}, s) = \left( \frac{1 - t_i + \frac{t_i \sigma_\theta^2}{2}}{1 - t_i} \right) v_\theta(t_i, A_{t_i}, s) - \left( \frac{t_i \sigma_\theta^2}{2(1 - t_i)\, t_i} \right) A_{t_i}, \tag{14}$$

with $b_\theta(0, \cdot, \cdot) = v_\theta(0, \cdot, \cdot)$. Intuitively, the first factor inflates the learned velocity to counteract diffusion, and the second term contracts towards the straight path so that the terminal law remains unchanged. The detailed proof can be found on pages 28–35 in (Holderrieth & Erives, 2025).

**Per-step transition.** Under Equ. (13), the conditional $A_{t_{i+1}} \mid A_{t_i}, s$ is Gaussian:

$$\eta_\theta\big(A_{t_{i+1}} \mid A_{t_i}, s; \Delta t_i\big) = \mathcal{N}\Big(A_{t_i} + b_\theta(t_i, A_{t_i}, s)\, \Delta t_i,\ \sigma_\theta^2\, \Delta t_i\, I_d\Big).$$

We denote $A_{t_0} \sim \mathcal{N}(0, I_d)$ as the base. The final action is $a = \tanh(A_{t_K})$.

## A.2  JOINT PATH DENSITY AND SQUASHING JACOBIAN

Let $\mathcal{A} = (A_{t_0}, \ldots, A_{t_K})$. The joint density factorizes as

$$p_c(\mathcal{A} \mid s) = \zeta(A_{t_0}) \prod_{i=0}^{K-1} \eta_\theta\big(A_{t_{i+1}} \mid A_{t_i}, s; \Delta t_i\big) \cdot \|\det \mathcal{J}(a)\|^{-1}, \quad a = \tanh(A_{t_K}), \tag{15}$$

where $\zeta$ is the standard Gaussian base density for $A_{t_0}$, $\eta_\theta(\cdot)$ is the per-step transition in Section A.1, and $\mathcal{J}(a)$ is the Jacobian of the element-wise $\tanh$ squashing. The marginal policy density follows by integrating out the intermediate pre-activations:

$$\pi_\theta(a \mid s) = \int \cdots \int p_c\big(A_{t_0}, \ldots, A_{t_{K-1}}, A_{t_K} = \tanh^{-1}(a) \mid s\big)\, dA_{t_0} \cdots dA_{t_{K-1}}. \tag{16}$$

For element-wise $\tanh$, $\|\det \mathcal{J}(a)\| = \prod_{j=1}^{d}(1 - a_j^2)^{-1}$.

## A.3  PATHWISE EXPANSION OF THE MARGINAL SCORE

We derive the gradient of $\mathbb{E}_a[\log \pi_\theta(a \mid s)]$. Using Equ. (15):

$$\nabla_\theta \mathbb{E}_a[\log \pi_\theta(a \mid s)] = \nabla_\theta \mathbb{E}_{\mathcal{A}}\left[\log \pi_\theta(a \mid s)\right]$$

$$= \nabla_\theta \mathbb{E}_{\mathcal{A}}\left[\log\left(\int \cdots \int p_c(A_{t_0}, \ldots, A_{t_{K-1}}, A_{t_K} = \tanh^{-1}(a) \mid s)\, dA_{t_0} \cdots dA_{t_{K-1}}\right)\right]. \tag{17}$$

Expanding the inner gradient yields

$$\nabla_\theta \log \pi_\theta(a \mid s) = \frac{1}{\pi_\theta(a \mid s)} \nabla_\theta \int \cdots \int \zeta(A_{t_0}) \left[ \prod_{i=0}^{K-1} \eta_\theta \left( A_{t_{i+1}} \mid A_{t_i}, s; \Delta t_i \right) \right] \| \det \mathcal{J}(a) \|^{-1} dA_{t_0:K-1}$$

$$= \frac{1}{\pi_\theta(a \mid s)} \int \cdots \int \zeta(A_{t_0}) \left[ \prod_{i=0}^{K-1} \eta_\theta \left( A_{t_{i+1}} \mid A_{t_i}, s; \Delta t_i \right) \right]$$

$$\cdot \sum_{i=0}^{K-1} \nabla_\theta \log \eta_\theta \left( A_{t_{i+1}} \mid A_{t_i}, s; \Delta t_i \right) \| \det \mathcal{J}(a) \|^{-1} dA_{t_0:K-1}. \tag{18}$$

Therefore,

$$\nabla_\theta \mathbb{E}_a[\log \pi_\theta(a \mid s)] = \mathbb{E}_\mathcal{A} \left[ \sum_{i=0}^{K-1} \nabla_\theta \log \eta_\theta \left( A_{t_{i+1}} \mid A_{t_i}, s; \Delta t_i \right) \right], \tag{19}$$

where the Jacobian term does not contribute because it is independent of $\theta$. Since $\eta_\theta$ is Gaussian with mean $m_i = A_{t_i} + b_\theta \Delta t_i$ and covariance $\Sigma_i = \sigma_\theta^2 \Delta t_i I$, each term is closed form:

$$\nabla_\theta \log \eta_\theta = \frac{1}{\sigma_\theta^2 \Delta t_i} \left( A_{t_{i+1}} - m_i \right)^\top \frac{\partial m_i}{\partial \theta} - \frac{d}{\sigma_\theta} \frac{\partial \sigma_\theta}{\partial \theta} + \text{higher-order terms if } \sigma_\theta \text{ depends on } \theta.$$

## A.4 Gradients of the SAC losses under the joint path factorization

**Critic update.** The target-matching loss is

$$L(\psi) = \left[ Q_\psi(s_h, a_h) - \left( r_h + \gamma Q_{\bar\psi}(s_{h+1}, a_{h+1}) - \alpha \log \pi_\theta(a_h \mid s_h) \right) \right]^2, \tag{20}$$

where $a_{h+1} \sim \pi_\theta(\cdot \mid s_{h+1})$. Using the joint-path form,

$$\nabla_\psi L(\psi) = 2 \left( Q_\psi(s_h, a_h) - \left( r_h + \gamma Q_{\bar\psi}(s_{h+1}, a_{h+1}) - \alpha \log p_c(\mathcal{A} \mid s) \right) \right) \nabla_\psi Q_\psi(s_h, a_h), \tag{21}$$

where no gradients flow through $Q_{\bar\psi}$. Replacing the marginal $\log \pi_\theta$ by $\log p_c$ only changes a baseline and has a negligible effect on learning behavior.

**Actor update.** The actor loss is

$$L(\theta) = \alpha \log \pi_\theta(a_h^\theta \mid s_h) - Q_\psi(s_h, a_h^\theta), \tag{22}$$

with $a_h^\theta = \tanh(A_{t_K}^\theta)$. Its gradient uses the pathwise form:

$$\nabla_\theta L(\theta) = \alpha \sum_{i=0}^{K-1} \nabla_\theta \log \eta_\theta \left( A_{t_{i+1}}^\theta \mid A_{t_i}^\theta, s_h; \Delta t_i \right) - \nabla_\theta Q_\psi(s_h, a_h^\theta), \tag{23}$$

where the $Q$-term differentiates through $a_h^\theta$.

## A.5 Path-Regularized Soft Critic

This section explains how the joint and marginal densities relate and why the resulting critic can be naturally interpreted as a path-regularized variant of maximum-entropy RL.

### A.5.1 Exact decomposition of the entropy term

For any $(s, a)$ with $a = \tanh(A_{t_K})$, the joint path-density admits the factorization

$$p_c(\mathcal{A} \mid s) = \pi_\theta(a \mid s) \, r_\theta(\mathcal{A} \mid a, s), \tag{24}$$

where $r_\theta(\mathcal{A} \mid a, s)$ is the conditional distribution of the latent path given the final action:

$$r_\theta(\mathcal{A} \mid a, s) = \frac{p_c(\mathcal{A} \mid s)}{\pi_\theta(a \mid s)}, \qquad \int r_\theta(\mathcal{A} \mid a, s) dA_{t_0:K-1} = 1. \tag{25}$$

Taking logarithms and averaging under $r_\theta(\cdot \mid a, s)$ yields the identity

$$\mathbb{E}_{r_\theta(\mathcal{A}|a,s)}\big[\log p_c(\mathcal{A} \mid s)\big] = \log \pi_\theta(a \mid s) + h(r_\theta(\cdot \mid a, s)), \tag{26}$$

where

$$h(r_\theta(\cdot \mid a, s)) = -\mathbb{E}_{r_\theta(\mathcal{A}|a,s)}[\log r_\theta(\mathcal{A} \mid a, s)] \tag{27}$$

is the differential entropy of the conditional path distribution.

Equation equation 26 shows that the expected surrogate penalty $-\log p_c(\mathcal{A} \mid s)$ differs from the true negative entropy $-\log \pi_\theta(a \mid s)$ by exactly the path-entropy term $h(r_\theta)$:

$$\mathbb{E}_{r_\theta}[-\log p_c(\mathcal{A} \mid s)] = -\log \pi_\theta(a \mid s) + h(r_\theta(\cdot \mid a, s)). \tag{28}$$

Identity equation 28 reveals that using the surrogate $-\log p_c(\mathcal{A} \mid s)$ in the critic corresponds to augmenting the original maximum-entropy objective with an additional path-entropy term:

$$J_{\text{ours}} = \mathbb{E}\Big[\sum_t r_t + \alpha \log \pi_\theta(a_t \mid s_t) + \alpha\, h(r_\theta(\cdot \mid a_t, s_t))\Big]. \tag{29}$$

The extra entropy term encourages the conditional path distribution $r_\theta(\cdot \mid a, s)$ to be diffuse rather than sharply concentrated. This induces a regularizing effect on the critic: actions whose flow rollouts exhibit high variability (large $h(r_\theta)$) receive an additional penalty through the surrogate. Empirically, this produces a more conservative soft $Q$-function and mitigates the well-known overestimation issues encountered in off-policy training.

### A.5.2 WHY THIS IS ACCEPTABLE FOR OFF-POLICY FLOW TRAINING

It is important to emphasize that the purpose of our method is not to exactly replicate the original SAC critic. Instead, our goal is to stabilize off-policy RL in the presence of multi-step flow rollouts, whose training is notoriously brittle due to compounding gradients and sensitivity to densityevaluation errors. The path-regularized critic trades a small, well-understood bias for substantially improved numerical stability. This trade-off is common and often desirable in deep RL, where exact Bellman equations are rarely satisfied under function approximation.

Moreover, the actor update in our method remains an exact policy gradient for the original maximum-entropy objective, thanks to the pathwise score expansion shown in Appendix A.3. The bias introduced by the critic therefore does not alter the policy objective being optimized; it only affects the value-based shaping signal used during training.

Finally, we note that the proposed flow-based policy construction (i.e., Flow-G and Flow-T) is not specific to SAC. Methods such as TD3 rely solely on $Q$-function targets and do not require evaluating log policy densities. In these settings, our flow policy can be used without any entropy-related complications. The critic design described above is only needed for maximum-entropy algorithms; other off-policy methods can directly adopt the same flow-based actor with no additional adjustments.

### A.6 TEMPERATURE UPDATE (LEARNED $\alpha$)

When learning $\alpha$ to match a target entropy $\bar{\mathcal{H}}$:

$$L(\alpha) = \mathbb{E}_{s_h, a_h^\theta \sim \pi_\theta(\cdot|s_h)}\left[-\alpha\left(\log \pi_\theta(a_h^\theta \mid s_h) + \bar{\mathcal{H}}\right)\right]. \tag{30}$$

The gradient is

$$\nabla_\alpha L(\alpha) = -\mathbb{E}_{s_h, a_h^\theta}\left[\log \pi_\theta(a_h^\theta \mid s_h) + \bar{\mathcal{H}}\right]. \tag{31}$$

Using the joint-path surrogate yields

$$\nabla_\alpha L(\alpha) = -\mathbb{E}\left[\sum_{i=0}^{K-1} \log \eta_\theta\left(A_{t_{i+1}}^\theta \mid A_{t_i}^\theta, s_h; \Delta t_i\right) - \log \|\det \mathcal{J}(a_h^\theta)\| + \bar{\mathcal{H}}\right], \tag{32}$$

and we set $\bar{\mathcal{H}} = 0$ unless otherwise noted.

A.7 PRACTICAL NOTES AND IMPLEMENTATION DETAILS

**Rollout length and noise.** Use small $K$ (e.g., $4$) to control backprop depth and latency. Fix $\sigma_\theta$ (e.g., $0.10$) or learn a lightweight state head; fixed schedules simplify tuning.

**Squashing and Jacobian.** Always squash $A_{t_K} \mapsto a = \tanh(A_{t_K})$ and include the exact Jacobian in $\log p_c$ of Equ. (15) to keep the entropy term correct.

**Targets and normalization.** Maintain a delayed target $Q_{\bar\psi}$ with EMA. Pre-normalization in Flow-T and a mild positive gate bias in Flow-G improve early stability.

**Gradient flow.** Flow-G gates the residual change to damp gradient amplification; Flow-T uses pre-norm residual blocks. Both act as drop-in $v_\theta$ inside Equ. (4).

**Offline-to-online.** In the regularized actor loss of the main text (Equ. (12)), choose $\beta$ large early to stay on-replay, then anneal as online data grows. Flow-matching pretraining via Equ. (3) is optional but helpful for sparse rewards.

**Efficiency.** The entropy term scales linearly in $K$ and action dimension $d$ because it decomposes into per-step Gaussian factors.

**Reproducibility.** We evaluate $\log p_c$ and its gradient with a single noisy rollout per update; additional variance reduction is possible but not required in our settings.

# B DETAILED ANALYSIS OF GRADIENT STABILITY

This section provides a more formal mathematical justification for the gradient pathologies in standard flow-based policies (when viewed as RNNs), as discussed in Section 3, and elaborates on how our Flow-G and Flow-T architectures address these issues.

## B.1 THE VANISHING/EXPLODING GRADIENT PROBLEM IN STANDARD FLOW ROLLOUTS

As established in Equation (5), the standard $K$-step flow rollout $A_{t_{i+1}} = A_{t_i} + \Delta t_i \, v_\theta(t_i, A_{t_i}, s)$ is algebraically equivalent to a residual RNN, where $A_{t_i}$ is the hidden state and $f_\theta(\cdot) = \Delta t_i \, v_\theta(\cdot)$ is the RNN cell.

In off-policy RL, the actor loss $L(\theta)$ is a function of the final action $A_{t_K}$ (e.g., $L(\theta) = -Q_\psi(s_h, \tanh(A_{t_K}^\theta))$ from Equation (10)). To update the parameters $\theta$ of the velocity network $v_\theta$, the gradient must be backpropagated through time (BPTT) from $A_{t_K}$ back to $A_{t_0}$.

Let us analyze the gradient flow. The gradient of the loss $L$ with respect to an intermediate state $A_{t_i}$ is:

$$\nabla_{A_{t_i}} L = (\nabla_{A_{t_K}} L) \cdot \prod_{j=i}^{K-1} \frac{\partial A_{t_{j+1}}}{\partial A_{t_j}}$$

where $\frac{\partial A_{t_{j+1}}}{\partial A_{t_j}}$ is the Jacobian matrix of the transition:

$$\frac{\partial A_{t_{j+1}}}{\partial A_{t_j}} = \frac{\partial}{\partial A_{t_j}} \left( A_{t_j} + f_\theta(t_j, A_{t_j}, s) \right) = I + \frac{\partial f_\theta(t_j, A_{t_j}, s)}{\partial A_{t_j}}$$

Substituting this back, the gradient propagation over $K$ steps becomes a long product of these Jacobians:

$$\nabla_{A_{t_0}} L = (\nabla_{A_{t_K}} L) \cdot \prod_{i=0}^{K-1} \left( I + \frac{\partial f_\theta(t_i, A_{t_i}, s)}{\partial A_{t_i}} \right) \tag{33}$$

This is the core problem. As identified in the seminal work on LSTM (Hochreiter & Schmidhuber, 1997), this long product of matrices is exponentially unstable. The error signal $\nabla_{A_{t_K}} L$ is repeatedly multiplied by the Jacobians of the state transition.

- **Exploding Gradients:** If the singular values of these Jacobians are persistently greater than 1, the norm of the gradient will grow exponentially, leading to unstable training, as seen in our Naive baseline in Fig. 6.

- **Vanishing Gradients:** Conversely, if the singular values are persistently less than 1, the norm of the gradient will shrink exponentially, preventing the error signal from $A_{t_K}$ from reaching the parameters that influenced $A_{t_0}$, $A_{t_1}$, etc. This makes learning long-term dependencies impossible.

This "gradient pathology" is the fundamental technical challenge that makes direct off-policy training of standard flow-based policies notoriously unstable.

## B.2 Flow-G and Flow-T as Gradient Stabilizers

To address the instability of Equation (33), the objective is to design an architecture $f_\theta$ such that the product of Jacobians remains well-conditioned, with singular values centered around 1.0. Our Flow-G and Flow-T designs are explicitly motivated by architectures from the sequence-modeling literature (LSTMs, GRUs, and Transformers) that were invented to solve this exact problem.

**The LSTM Precedent:**   The key innovation of LSTM (Hochreiter & Schmidhuber, 1997) was the Constant Error Carousel (CEC). LSTM introduced the multiplicative gates (input, forget, output), which are then trained to learn when to allow error signals into this stable "carousel" and when to use the information stored within it.

Our Flow-G (GRU) and Flow-T (Transformer) architectures achieve a similar outcome through related, albeit more complex, mechanisms.

**Flow-G (GRU-gated):**   The Flow-G velocity (Equation 6) is $v_\theta = g_i \odot (\hat{v}_\theta - A_{t_i})$. The rollout step becomes:

$$A_{t_{i+1}} = A_{t_i} + \Delta t_i \left( g_i \odot (\hat{v}_\theta(t_i, A_{t_i}, s) - A_{t_i}) \right)$$

Rewriting this per-dimension (with $g_i^{(d)}$ being the $d$-th dimension of the gate):

$$A_{t_{i+1}}^{(d)} = \left( 1 - \Delta t_i g_i^{(d)} \right) A_{t_i}^{(d)} + \Delta t_i g_i^{(d)} \hat{v}_\theta^{(d)}(\cdot)$$

This is precisely the update form of a Gated Recurrent Unit (GRU). The Jacobian of this transition (ignoring terms from $\partial g_i / \partial A_{t_i}$ and $\partial \hat{v}_\theta / \partial A_{t_i}$ for clarity) is approximately:

$$\frac{\partial A_{t_{i+1}}}{\partial A_{t_i}} \approx I - \text{diag}(\Delta t_i \, g_i)$$

The crucial insight is that the gate $g_i \in [0, 1]$ is a learnable parameter. If the network needs to preserve information (and its gradient) across many steps, it can learn to set $g_i \to 0$ for those steps. When $g_i \to 0$, the Jacobian $\frac{\partial A_{t_{i+1}}}{\partial A_{t_i}} \to I$. This mimics the CEC, allowing the gradient to flow unimpeded. Flow-G thus learns to dynamically regulate its own gradient stability, just as a GRU or LSTM does.

**Flow-T (Transformer-decoded):**   The Flow-T architecture (Equations 8-9) achieves stability not through explicit gating, but through its architectural design, which is standard in modern Transformers. The Jacobian is $\frac{\partial A_{t_{i+1}}}{\partial A_{t_i}} = I + \Delta t_i \frac{\partial v_\theta}{\partial A_{t_i}}$. Stability hinges on ensuring the Jacobian of the velocity network, $\frac{\partial v_\theta}{\partial A_{t_i}}$, is well-behaved.

Flow-T accomplishes this via two key components:

1. **Pre-Layer Normalization (Pre-LN):** As shown in Equation (8), all inputs to the Cross-Attention and FFN sub-layers are passed through Layer Normalization ($\text{LN}(\cdot)$). Pre-LN ensures the inputs to each layer are normalized, which has been shown to bound the magnitude of activations and their gradients, leading to a much more stable and "well-behaved" loss landscape.

2. **Residual Connections:** The outputs of the Cross-Attention and FFN blocks are added to their inputs. This residual stream, ubiquitous in modern deep learning, provides a clean identity path for gradients to flow backward, bypassing the complex computations of the sub-layers.

This combination of Pre-LN and residual connections is a cornerstone of modern Transformer architectures precisely because it stabilizes gradients in very deep networks. It ensures the spectral norm of the velocity Jacobian, $||\frac{\partial v_\theta}{\partial A_{t_i}}||$, remains controlled. This achieves the same goal as the CEC: it keeps the overall step Jacobian $\frac{\partial A_{t_{i+1}}}{\partial A_{t_i}}$ close to the identity matrix, preventing the product in Equation (33) from exploding or vanishing.

**A Necessary Caveat:** We must emphasize that unlike the original LSTM's Jocobian $\frac{\partial c_t}{\partial c_{t-1}}$, which provides a provable guarantee of $\frac{\partial c_t}{\partial c_{t-1}} = I$ (in its simplest form), a similar exact proof for Flow-G and Flow-T is intractable. The non-linear complexity of the candidate networks ($\hat{v}_\theta$), the gates ($g_i$ which also depend on $A_{t_i}$), and the multi-layer, multi-head attention blocks (in Flow-T) makes a closed-form analysis of the full Jacobian product (Equation 33) infeasible.

However, our designs are not arbitrary. By importing these specific architectural motifs—which were explicitly engineered in the sequence-modeling literature to solve the vanishing/exploding gradient problem—we create a strong inductive bias towards gradient stability. Our architectures effectively approximate a constant-norm gradient path, which is what enables stable end-to-end off-policy optimization, as our empirical results in Figure 6 and our ablation studies robustly confirm.

## C    EXTENDED RELATED WORK

We evaluate our approach against several state-of-the-art methods, categorized into two groups based on their training paradigm. From-scratch algorithms initialize randomly and learn entirely through environment interaction, while offline-to-online methods first pre-train on expert demonstrations before transitioning to online reinforcement learning.

### C.1    FROM-SCRATCH TRAINING METHODS

The integration of generative models into reinforcement learning has emerged as a prominent research direction, with particular focus on training policies parameterized by diffusion and flow-based models. This line of work addresses the limitations of traditional unimodal policy representations by leveraging the expressive power of generative models to capture complex, multimodal action distributions.

Early efforts in this domain primarily concentrated on diffusion-based policies. Q-Score Matching (QSM) (Psenka et al., 2024) pioneered this direction by establishing a theoretical connection between score functions and Q-value gradients, enabling direct policy optimization through score matching objectives. Building upon this foundation, several advanced methods have been proposed: QVPO (Ding et al., 2024) introduces Q-weighted variational policy optimization for improved sample efficiency; DDiffPG (Li et al., 2024) extends policy gradient methods to diffusion models; MaxEntDP (Dong et al., 2025) incorporates maximum entropy principles; and DIME (Celik et al., 2025) reformulates diffusion policy training through KL divergence minimization between denoising chains and exponentiated critic targets.

More recently, attention has shifted toward flow-based policies, which offer computational advantages over diffusion models through deterministic ODE integration. FlowRL (Lv et al., 2025) represents the current state-of-the-art in this category, proposing Wasserstein-2 regularized policy optimization that constrains the learned policy to remain within proximity of optimal behaviors identified in the replay buffer.

For our experimental evaluation, we select DIME and FlowRL as primary benchmarks for diffusion and flow-based approaches, respectively, based on their reported performance improvements over earlier methods such as QVPO and QSM. We additionally include QSM in our comparison as it established many of the foundational concepts underlying subsequent developments in this field. Meanwhile, classical RL training methods for Gaussian policy, including PPO (Schulman et al., 2017) and SAC (Haarnoja et al., 2018).

**FlowRL (Lv et al., 2025).** This approach directly optimizes flow-based policies using off-policy RL with Wasserstein regularization. The critic $Q_\psi(s, a)$ follows standard SAC updates, minimizing

the temporal difference error:

$$L_Q(\psi) = \mathbb{E}_{(s,a,r,s')\sim\mathcal{D}}\left[(Q_\psi(s,a) - (r + \gamma\mathbb{E}_{a'\sim\pi_\theta}[Q_\psi(s',a')]))^2\right]. \tag{34}$$

The key innovation lies in the actor update, which formulates policy optimization as a constrained problem that maximizes Q-values while regularizing the velocity field $v_\theta$ using a Wasserstein-2 distance constraint. In practice, this is solved using a Lagrangian relaxation:

$$L_\pi(\theta) = \mathbb{E}_{\substack{s,a\sim\mathcal{D},a'\sim\pi_\theta \\ t\sim U(0,1)}}\left[f(Q_{\pi_{\beta*}}(s,a) - Q_\psi(s,a'))\|v_\theta(s,A_t,t) - (a-a_0)\|^2\right], \tag{35}$$

where $f(\cdot)$ is a non-negative weighting function, $A_t = (1-t)a_0 + ta$ represents the flow interpolation path, and $\pi_{\beta*}$ denotes the optimal behavior policy derived from the replay buffer. The constraint adaptively regularizes the policy toward high-performing behaviors when $Q_{\pi_{\beta*}} > Q_\psi$, effectively aligning the flow optimization with value-based policy improvement.

**DIME (Celik et al., 2025).** This method treats diffusion policies as exponential family distributions and optimizes them via KL divergence minimization. The critic update remains standard:

$$L_Q(\psi) = \frac{1}{2}\mathbb{E}\left[(Q_\psi(s_t,a_t) - Q_{\text{target}}(s_t,a_t))^2\right]. \tag{36}$$

The actor update is more sophisticated, defining a target marginal through the exponentiated critic $\bar{\pi}_0(a|s) = \exp(Q_\psi(s,a))/Z_\psi(s)$ and minimizing the KL divergence between the denoising chain and this target:

$$L(\theta) = \mathbb{E}_{\pi_\theta}\left[\log\pi_\theta(a_N|s) - Q_\psi(s,a_0) + \sum_{n=1}^{N}\log\frac{\pi_\theta(a_{n-1}|a_n,s)}{\bar{\pi}(a_n|a_{n-1},s)}\right]. \tag{37}$$

**QSM (Q-Score Matching) (Psenka et al., 2024)** This approach leverages score matching to align the policy's score function with the action gradient of the Q-function, providing a principled connection between value-based and score-based learning. The critic follows a double Q-learning update with target networks for stability:

$$L_Q(\theta) = \mathbb{E}_{(s_t,a_t,r_{t+1},s_{t+1})\sim\mathcal{B}}\left[\left(Q_\theta(s_t,a_t) - \left(r_{t+1} + \gamma\min_{i=1,2}Q_{\theta_i'}(s_{t+1},a_{t+1})\right)\right)^2\right], \tag{38}$$

where $Q_{\theta_i'}$ denotes the target networks. The actor update represents the core innovation, training a score function $\Psi_\phi(s_t,a_t)$ to match the scaled action gradient of the Q-function:

$$L_\pi(\phi) = \mathbb{E}_{(s_t,a_t)\sim\mathcal{B}}\left[\|\Psi_\phi(s_t,a_t) - \alpha\nabla_a Q_\theta(s_t,a_t)\|^2\right], \tag{39}$$

where $\alpha$ controls the alignment strength. This formulation enables the policy to naturally follow the Q-function's action gradients, providing implicit policy improvement without explicit action sampling.

## C.2 OFFLINE-TO-ONLINE TRAINING METHODS

While from-scratch training is viable for many reinforcement learning tasks, it often struggles with sample efficiency in complex environments, particularly those with dense rewards. To address this limitation, the offline-to-online paradigm has become a prominent approach. This strategy involves two stages: first, pre-training a policy on an offline dataset of expert behaviors, and second, fine-tuning this policy through online interaction with the environment.

This paradigm was initially explored with diffusion-based policies, leading to the development of methods such as DPPO (Ren et al., 2024), D3P (Yu et al., 2025), Resip (Ankile et al., 2025), and PA-RL (Mark et al., 2024). More recently, research has extended this approach to flow-based policies, which are the focus of our work.

Within the flow-policy literature, methods can be categorized by their online fine-tuning algorithm. For on-policy fine-tuning, ReinFlow (Zhang et al., 2025) stands out by successfully adapting a pre-trained flow-based policy using the PPO algorithm. For off-policy fine-tuning, FQL (Park et al.,

2025) and its successor QC-FQL (Li et al., 2025) are state-of-the-art. However, a crucial characteristic of these off-policy methods is their reliance on an auxiliary, distilled policy for online updates; they do not directly fine-tune the original flow model. Instead, they distill knowledge from the pre-trained flow-based policy into a simpler, unimodal policy that is more amenable to traditional off-policy RL.

For our experiments, we select ReinFlow, FQL, and QC-FQL as benchmarks. Our evaluation primarily concentrates on the off-policy methods to demonstrate the effectiveness of our proposed direct fine-tuning approach for flow-based policies.

**QC-FQL (Li et al., 2025)**    This approach employs a three-network architecture: a critic $Q_\theta$, a one-step noise-conditioned policy $\mu_\psi$, and a behavior flow-based policy $f_\xi$. The method extends FQL to handle action chunking by operating in temporally extended action spaces. The critic processes action sequences and is updated via:

$$L_Q(\theta_k) = \left( Q_{\theta_k}(s_t, a_t, \ldots, a_{t+h-1}) - r_t^h - \frac{1}{K} \sum_{k'=1}^{K} Q_{\bar{\theta}_{k'}}(s_{t+h}, a_{t+h}, \ldots, a_{t+2h-1}) \right)^2, \quad (40)$$

where $r_t^h$ represents the cumulative discounted reward over the action chunk horizon. The one-step policy is trained to maximize Q-values while maintaining proximity to the behavior policy outputs:

$$L_\pi(\psi) = -Q_\theta(s_t, \mu_\psi(s_t, z_t)) + \alpha \left\| \mu_\psi(s_t, z_t) - \left[ a_t^\xi \cdots a_{t+h-1}^\xi \right] \right\|_2^2. \quad (41)$$

**FQL (Park et al., 2025).**    This method represents a simplified version of QC-FQL with unit action chunks ($h = 1$). The critic follows standard Bellman updates while the actor combines value maximization with distillation regularization. The actor loss explicitly balances Q-value optimization against behavioral constraints:

$$L_\pi(\omega) = \mathbb{E}_{s \sim \mathcal{D}, a^\pi \sim \mu_\omega}[-Q_\phi(s, a^\pi)] + \alpha \mathbb{E}_{s \sim \mathcal{D}, z \sim \mathcal{N}(0,I)} \left[ \|\mu_\omega(s, z) - \mu_\theta(s, z)\|_2^2 \right], \quad (42)$$

where $\mu_\theta$ represents a pre-trained behavioral clone used for regularization.

**ReinFlow (Zhang et al., 2025)**    This approach augments flow-based policies with noise injection networks to enable efficient likelihood computation during policy gradient updates. Following a warm-up phase for critic training, the method jointly optimizes the flow-based policy $\pi_\theta$ and noise injection network $\sigma_{\theta'}$ through:

$$L_\pi(\theta, \theta') = \mathbb{E} \left[ -A_{\theta_{\text{old}}}(s, a) \sum_{k=0}^{K-1} \log \pi_\theta(a_{k+1}|a_k, s) + \alpha \cdot R(a, s; \theta, \theta_{\text{old}}) \right], \quad (43)$$

where $A_{\theta_{\text{old}}}(s, a)$ denotes advantage estimates and $R(\cdot)$ provides regularization to prevent excessive deviation from the previous policy.

**Key Distinctions.**    Unlike these baseline approaches, our method enables direct training the flow-based policy via SAC (off-policy methods) without requiring auxiliary distillation actors, surrogate objectives, or complex multi-network architectures. The Flow-G and Flow-T parameterizations provide gradient stability while maintaining the expressive power of the original flow-based policy throughout training.

## D    EXPERIMENTAL DOMAIN

To comprehensively evaluate our method, we conduct experiments across a diverse suite of simulated environments. We utilize the classic **MuJoCo** benchmark (Todorov et al., 2012) for standard from-scratch reinforcement learning. To assess performance in the more challenging offline-to-online setting, particularly with sparse rewards, we employ complex manipulation tasks from **OG-Bench** (Park et al., 2024) and human-demonstration-based tasks from **Robomimic** (Mandlekar et al., 2021). Visualizations of these environments are presented in Fig. 8.

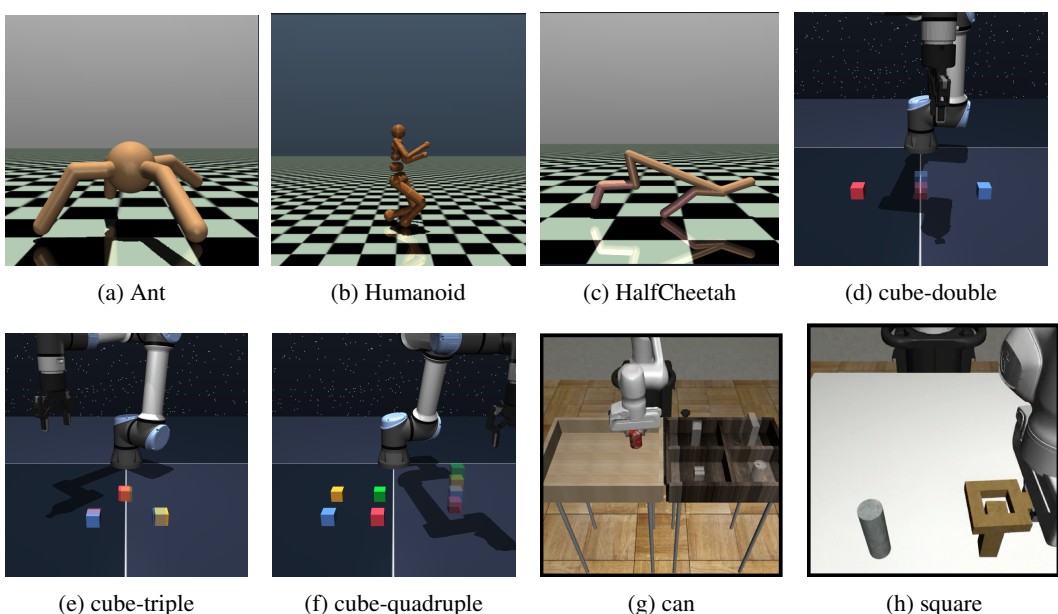

(a) Ant      (b) Humanoid      (c) HalfCheetah      (d) cube-double

(e) cube-triple      (f) cube-quadruple      (g) can      (h) square

Figure 8: Visualizations of the diverse simulation environments used for evaluation. Subfigures (a-c) show the **MuJoCo** locomotion tasks. Subfigures (d-f) depict the complex, sparse-reward manipulation tasks from **OGBench**. Subfigures (g-h) illustrate the demonstration-based tasks from **Robomimic**. This selection provides a comprehensive testbed for evaluating both from-scratch learning and offline-to-online fine-tuning.

### D.1 MUJOCO ENVIRONMENTS

We evaluate our method on six standard continuous control tasks from the **MuJoCo** physics simulation benchmark (Todorov et al., 2012): `Hopper-v4`, `Walker2d-v4`, `HalfCheetah-v4`, `Ant-v4`, `Humanoid-v4`, and `HumanoidStandup-v4`. These environments feature simulated robots with varying degrees of complexity, where the primary objective is to learn a locomotion policy that maximizes forward velocity without falling. They serve as a standard measure of performance for from-scratch RL algorithms.

### D.2 OGBENCH ENVIRONMENTS

From **OGBench** (Park et al., 2024), we select four challenging manipulation domains using their publicly available single-task versions. The selected domains include `cube-double/triple/quadruple` tasks. In the cube tasks, an agent must control a UR-5 arm to place multiple objects in target locations, receiving a reward of $-n_{\text{wrong}}$, where $n_{\text{wrong}}$ is the number of incorrectly placed cubes. The cube-triple and cube-quadruple tasks are particularly difficult to solve from offline data alone, providing a rigorous testbed for the sample efficiency of offline-to-online algorithms. In the offline phase, we use the official 100M-size dataset[2].

### D.3 ROBOMIMIC ENVIRONMENTS

We use three robotic manipulation tasks from the **Robomimic** benchmark (Mandlekar et al., 2021), utilizing the multi-human datasets which contain 300 successful demonstration trajectories per task. The tasks are selected to represent a range of difficulties: `Lift`, a simple pick-and-place task involving a cube; `Can`, an intermediate task requiring placing a can into a bin; and `Square`, the most challenging task, which requires the precise insertion of a square nut onto a peg. We use the official Multi-Human (MH) dataset, containing 300 mixed trajectories per task, for offline pre-training.

---

[2]`https://github.com/seohongpark/ogbench?tab=readme-ov-file`

Table 1: Details of the experimental environments. The tasks span classic continuous control with dense rewards (MuJoCo), complex manipulation with sparse rewards (OGBench), and challenging imitation-based tasks also framed with sparse rewards (Robomimic). This selection provides a comprehensive benchmark with diverse state spaces, action dimensions, and reward structures. We use the same dataset configuration in (Li et al., 2025).

| Tasks | Reward Type | Dataset Size | Episode Length | Action Dimension |
|---|---|---|---|---|
| *MuJoCo* | | | | |
| Hopper-v4 | Dense | / | 1000 | 3 |
| Walker2d-v4 | Dense | / | 1000 | 6 |
| HalfCheetah-v4 | Dense | / | 1000 | 6 |
| Ant-v4 | Dense | / | 1000 | 8 |
| Humanoid-v4 | Dense | / | 1000 | 17 |
| HumanoidStandup-v4 | Dense | / | 1000 | 17 |
| *OGBench* | | | | |
| cube-double | Sparse | 1M | 500 | 5 |
| cube-triple | Sparse | 3M | 1000 | 5 |
| cube-quadruple-100M | Sparse | 100M | 1000 | 5 |
| *Robomimic* | | | | |
| lift | Sparse | 31,127 | 500 | 7 |
| can | Sparse | 62,756 | 500 | 7 |
| square | Sparse | 80,731 | 500 | 7 |

# E   IMPLEMENTATION DETAILS FOR EXPERIMENTS

In this section, we introduce the implementation details of the hyperparameter setting and network structures. We first begin with the from-scratch training:

## E.1   FROM-SCRATCH TRAINING SETTING

In from-scratch training, we develop our algorithm based on CleanRL (Huang et al., 2022), which is a widely used benchmark codebase, where we also use the same implementation of PPO, SAC in it. The hyperparameters for SAC, PPO, and DIME are available in Tab. 2, 3, and 4. For FlowRL , we use the official implementation except for unifying the parameter quantity. We run 5 seeds for all experiments and all plots use a 95% confidenceinterval.

Table 2: Common Hyperparameters for SAC Algorithms

| Parameter | Value |
|---|---|
| Optimizer | Adam |
| | ($b_1 = 0.5$ for Flow-based approaches) |
| Batch size ($M$) | 512 |
| Replay buffer size | $1 \times 10^6$ |
| Discount factor ($\gamma$) | 0.99 |
| Policy learning rate | $3 \times 10^{-4}$ |
| Critic learning rate | $1 \times 10^{-3}$ |
| Target network update rate ($\tau$) | 0.005 for **SAC** |
| | 1.0 for **Flow**, **Flow-G**, **Flow-T** |
| Learning starts | 50,000 |
| Entropy coefficient ($\alpha$) | 0.2 (initial value) |
| Target entropy | $-dim(\mathcal{A})$ for **SAC**, 0 for **Flow**, **Flow-G**, **Flow-T** |
| Automatic entropy tuning | True |
| Number of online environment steps | $1 \times 10^6$ |

Table 3: Hyperparameters for PPO

| Parameter | Value |
|---|---|
| Optimizer | Adam |
| Parallel envs | 32 |
| Discount factor ($\gamma$) | 0.99 |
| GAE lambda | 0.95 |
| Learning rate | $6 \times 10^{-4}$ |
| Num steps | 1024 |
| Num minibatches | 1 |
| Update epochs | 10 |
| Max grad norm | 10.0 |
| Clip coefficient ($\epsilon$) | 0.2 |
| Entropy coefficient | 0.01 |
| Total Timesteps | $1 \times 10^7$ |

Table 4: Hyperparameters for DIME

| Parameter | Value |
|---|---|
| Discount factor ($\gamma$) | 0.99 |
| Target network update rate ($\tau$) | 1.0 |
| Policy $tau$ | 1.0 |
| UTD | 1 |
| Policy delay | 3 |
| Batch size | 512 |
| Critic v_min | -1600 |
| Critic v_max | 1600 |
| Actor lr | $3 \times 10^{-4}$ |
| Critic lr | $3 \times 10^{-4}$ |
| Entropy coefficient ($\alpha$) | 1.0 (init) |
| Target entropy | 6.0 |
| Total Timesteps | $1 \times 10^6$ |

**Architectures of the velocity network in flow-based policies (Figs. 9–11).** We detail the network parameterizations for the velocity field $v_\theta$ used inside the flow rollout in Equ. (4). Across all variants, the policy starts from a state-conditioned base $A_{t_0} \sim \mathcal{N}(0, I_d)$, performs $K$ Euler updates $A_{t_{i+1}} = A_{t_i} + \Delta t_i\, v_\theta(t_i, A_{t_i}, s)$, and then applies $\tanh$ squashing to obtain the final action $a = \tanh(A_{t_K})$. During training, we optionally pair $v_\theta$ with a log-standard-deviation head to define the per-step Gaussian transition factors used by our noisy/likelihood-friendly rollout.

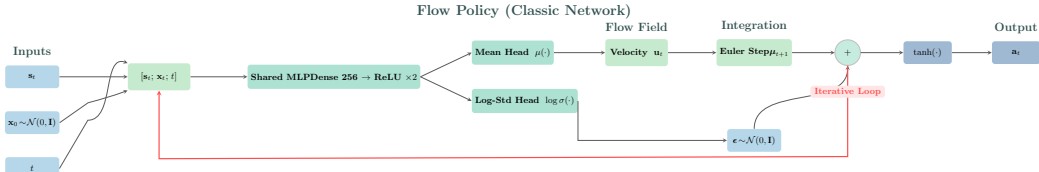

Figure 9: The flow-based policy designed with classic networks. The velocity is modeled with an arbitrary network; here, we use an MLP as the representative. The whole flow rollout corresponds to the recurrent computation of the residual RNN.

**Fig. 9: Classic (MLP) velocity network.** The baseline flow-based policy instantiates $v_\theta$ with a feed-forward network that is conditioned on the current intermediate action $A_{t_i}$, the environment state $s$, and the normalized time index $t_i$. Concretely, the input token is the concatenation $[\, s;\, A_{t_i};\, t_i\,]$, followed by a shared MLP trunk and two small heads: (i) a mean head $\mu_\theta(\cdot)$ that produces the deterministic velocity

$$v_\theta(t_i, A_{t_i}, s) = \mu_\theta([s; A_{t_i}; t_i]),$$

and (ii) a log-standard-deviation head $\log \sigma_\theta(\cdot)$ that parameterizes the per-step transition variance when we use the noisy rollout for likelihood-based training. Plugging this $v_\theta$ into Equ. (4) yields the standard residual update $A_{t_{i+1}} = A_{t_i} + \Delta t_i\, \mu_\theta([s; A_{t_i}; t_i])$. Algebraically, this is a residual RNN step with residual function $f_\theta(\cdot) = \Delta t_i\, \mu_\theta(\cdot)$, matching our sequence-model view in Equ. (5).

**Fig. 10: Gated velocity (Flow-G).** To stabilize gradients across the $K$ sampling steps, we replace the plain MLP velocity with a GRU-style gated update. Let $f_z$ (gate network) and $f_h$ (candidate network) be two MLPs taking $[s; A_{t_i}; t_i]$ as input. Define the update gate and the candidate as

$$g_i = \sigma(f_z([s; A_{t_i}; t_i])), \qquad \hat{v}_\theta = \phi(f_h([s; A_{t_i}; t_i])),$$

where $\sigma(\cdot)$ is the logistic sigmoid and $\phi(\cdot)$ is a bounded activation (e.g., $\tanh$). The gated velocity is then

$$v_\theta(t_i, A_{t_i}, s) = g_i \odot (\hat{v}_\theta - A_{t_i}),$$

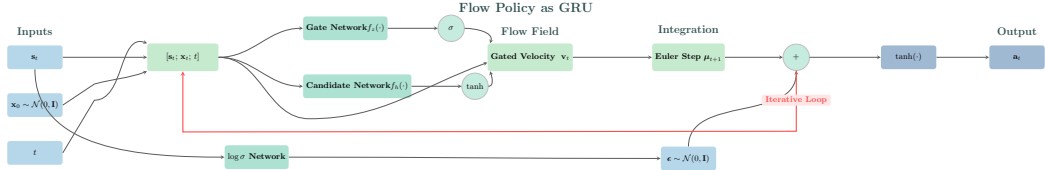

Figure 10: The flow-based policy designed with Gated velocity. The velocity is modeled with both the gate network and the candidate network. The whole flow rollout corresponds to the recurrent computation of GRU.

which, when inserted into Equ. (4), yields the GRU-like residual step

$$A_{t_{i+1}} = A_{t_i} + \Delta t_i \big(g_i \odot (\hat{v}_\theta - A_{t_i})\big),$$

exactly as in Equ. (6). Intuitively, $g_i$ interpolates between "keeping" the current intermediate action ($g_i \approx 0$) and "rewriting" it by the candidate proposal ($g_i \approx 1$). As in Fig. 9, we also include a $\log \sigma_\theta(\cdot)$ head for the per-step Gaussian factors used by the noisy rollout.

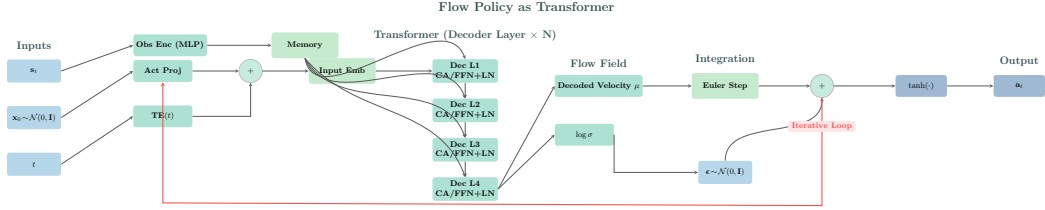

Figure 11: A schematic .

**Fig. 11: Transformer-decoder velocity (Flow-T).** Here we implement $v_\theta$ with a Transformer-style, pre-norm residual block that conditions on the state through cross-attention. We first form separate embeddings for the action-time token and the state:

$$\Phi_{A_i} = E_A\big(\phi_t(t_i), A_{t_i}\big), \qquad \Phi_S = E_S\big(\phi_s(s)\big),$$

as in Equ. (7), where $E_A, E_S$ are linear projections and $\phi_t, \phi_s$ are positional/feature encoders. We stack $L$ ($L = 4$ in this figure) pre-norm decoder blocks. In each layer $l = 1, \ldots, L$, the action token is refined by a state-only cross-attention and a feed-forward network (no token-to-token mixing across time positions):

$$Y_i^{(l)} = \Phi_{A_i}^{(l-1)} + \text{Cross}_l\Big(\text{LN}\big(\Phi_{A_i}^{(l-1)}\big), \text{context} = \text{LN}(\Phi_S)\Big), \qquad \Phi_{A_i}^{(l)} = Y_i^{(l)} + \text{FFN}_l\big(\text{LN}(Y_i^{(l)})\big),$$

as Equ. (8). Finally, the decoded token is projected to the velocity space

$$v_\theta(t_i, A_{t_i}, s) = W_o\big(\text{LN}(\Phi_{A_i}^{(L)})\big),$$

and the rollout step follows Equ. (4):

$$A_{t_{i+1}} = A_{t_i} + \Delta t_i \, W_o\big(\text{LN}(\Phi_{A_i}^{(L)})\big),$$

which matches Equ. (9). As in the other variants, a parallel $\log \sigma_\theta(\cdot)$ head provides per-step variances for the Gaussian transition factors.

**Takeaway: mapping to sequential models.** Under our sequence-model perspective, the classic MLP velocity in Fig. 9 realizes a residual RNN step in Equ. (5), the gated velocity in Fig. 10 realizes a GRU-style residual update in Equ. (6), and the decoded velocity in Fig. 11 realizes a Transformer Decoder refinement in Equ. (7)–(9). All three are drop-in parameterizations of $v_\theta$ inside the same flow rollout in Equ. (4), differing only in how they regulate and condition information flow across rollout steps.

Table 5: Actor (velocity) architectures inside the $K$-step flow rollout $A_{t_{i+1}} = A_{t_i} + \Delta t\, v_\theta(t_i, A_{t_i}, s)$. All variants apply `tanh` squashing with Jacobian correction. Notation: $d_a := |\mathcal{A}|$, Transformer $d=64$, heads $n_H=4$, layers $n_L=2$.

| Aspect | Classic (MLP) | Flow-G (GRU-gated) | Flow-T (Transformer-decoder) |
|---|---|---|---|
| Conditioning input | $[s;\ A_{t_i};\ t_i]$ | $[s;\ A_{t_i};\ \text{time\_emb}(t_i)]$ | $A_{t_i}$ token + time emb
state $s$ as memory |
| Backbone / blocks | MLP $256 \to 256$
ReLU | Gate: $128 \to d_a$ (`swish`)
Cand: $128 \to d_a$ (`swish`) | Decoder $\times n_L=2$
self-only SA, cross-attn($s$), FFN $4d$, LN |
| Velocity form | $v_\theta = \mu_\theta([s; A_{t_i}; t_i])$
$(\mu_\theta \in \mathbb{R}^{d_a})$ | $g_i = \sigma(f_z),\ \hat{v}=50\,\tanh(f_h)$
$v_\theta = g_i \odot (\hat{v} - A_{t_i})$ | $z_i = W_o(\text{LN}(\Phi^{(L)}_{A_i}))$
$v_\theta = z_i$ |
| Log-std clamp | tanh to
$[-5, 2]$ | tanh to
$[-5, 2]$ | tanh to
$[-5, 2]$ |
| Action sampling steps $K$ | 4 | 4 | 4 |
| Notable inits / dims | – | Gate head init: $W=0, b=5.0$
hidden 128 | $d=64, n_H=4, n_L=2$
obs-enc $32 \xrightarrow{\text{silu}} 64$ |
| Per-step update | $A \leftarrow A + v\,\Delta t$
$A \leftarrow \mathcal{N}(A, \sigma^2)$ | Same as Classic | Same as Classic |

## E.2 OFFLINE-TO-ONLINE TRAINING SETTING

The network design in offline-to-online training is similar to the from-scratch training. Recall the actor loss:

$$L(\theta) = \alpha \log p_c(\mathcal{A}^\theta \mid s_h) - Q_\psi(s_h, a_h^\theta) + \beta \|a_h^\theta - a_h\|^2, \quad (a_h, s_h) \sim \mathcal{B}.$$

It is observed that the setting of hyper-parameter $\beta$ highly influences the training, where the regularization decides whether the optimized policy stays close to or not to the policy in the buffer. We basically adopt the same setting of $\beta$ as (Li et al., 2025), where we detail in the Table 6:

Table 6: A comparison of the regularization parameter $\beta$ across different environments and algorithms. The notation $a/b$ specifies the value of the regularization parameter $\beta$ for the offline learning phase ($a$) and the subsequent online learning phase ($b$). For instance, $10000/1000$ indicates that $\beta = 10000$ is used for offline training and $\beta = 1000$ for online training.

| Environments | FQL | QC-FQL | Flow-G | Flow-T |
|---|---|---|---|---|
| scene-sparse-* | 300 | 300 | 300 | 300 |
| cube-double-* | 300 | 300 | 300 | 300 |
| cube-triple-* | 300 | 100 | 100 | 100 |
| cube-quadruple-100M-* | 300 | 100 | 100 | 100 |
| lift | 10000 | 10000 | 10000/1000 | 10000/1000 |
| can | 10000 | 10000 | 10000/1000 | 10000/1000 |
| square | 10000 | 10000 | 10000/1000 | 10000/1000 |

Table 7 summarizes the actor-side architectures and hyperparameters for our offline-to-online variants. We adopt action chunking (horizon $H$), which has been shown to be effective on complex tasks (Li et al., 2025). The parameter counts of Flow-G and Flow-T are less than or comparable to that of QC-FQL. We also use fewer denoising/sampling steps $K$ than QC-FQL to improve efficiency without degrading training quality. For stability, we set the SAC target entropy to 0 and employ a fixed sampling noise level—contrary to our from-scratch setting, where a separate network adaptively tunes the noise schedule.

**Sampling Steps Justification.** We justify our choice of sampling steps ($K = 4$ for our flow policies, $K = 16$ for diffusion baselines) based on two factors. First, this reflects the inherent efficiency of the models, as flow-based models (especially Rectified Flow) generally require significantly fewer integration steps than diffusion models require for denoising. Second, and most importantly, we

Table 7: Offline-to-online settings and actor-specific hyperparameters.

| Aspect | QC-FQL | Flow-G | Flow-T |
|---|---|---|---|
| Actor backbone | MLP ($512{\times}4$) GELU, no LN | MLP ($512{\times}4$) + gate ($h{=}256$, `swish`) | Decoder $n_L{=}2$, $d{=}128$ $n_H{=}4$, FFN $4d$ |
| Velocity form | $v_\theta(s, a, t)$ by MLP | $v{=}z \odot (50\tanh(\hat{v}) - a)$ $z{=}\sigma(f_z)$ | $v$ from decoder head (self+cross attn) |
| Flow / steps | Action sampling steps $K{=}10$ | Action sampling steps $K{=}4$ | Action sampling steps $K{=}4$ |
| Sampling noise std | deterministic | 0.10 | 0.10 |
| SAC entropy ($\alpha$) | N/A (no SAC) | autotune (init 0.2), $\alpha_{lr}{=}3{\times}10^{-4}$, $\bar{\mathcal{H}}{=}0$ | autotune (init 0.2), $\alpha_{lr}{=}3{\times}10^{-4}$, $\bar{\mathcal{H}}{=}0$ |
| Action range | `tanh` squash (deterministic) | `tanh` + Jacobian (for log-prob) | `tanh` + Jacobian (for log-prob) |
| Gate init / dims | — | gate head: $W{=}0$, $b{=}5.0$ hidden 256 | — |
| Transformer dims | — | — | $d{=}128$, $n_H{=}4$, $n_L{=}2$ |
| Actor hidden dims | $(512, 512, 512, 512)$ | $(512, 512, 512, 512)$ | (used only in enc./FFN; decoder per row above) |
| Action chunking | `True` | `True` | `True` |
| Opt / LR / WD | Adam, $3{\times}10^{-4}$ | Adam, $3{\times}10^{-4}$ | Adam, $3{\times}10^{-4}$ |
| Batch / $\gamma$ / $\tau$ | 256 / 0.99 / 0.005 | 256 / 0.99 / 0.005 | 256 / 0.99 / 0.005 |

adopted these values to ensure a fair and direct comparison with the key baseline papers. Our use of 4 sampling steps for SAC Flow-G and SAC Flow-T follows the established setting in the Rein-Flow (Zhang et al., 2025) baseline. Similarly, our use of 16 denoising steps for the diffusion-based baselines (e.g., DIME) matches the hyperparameter used in the DIME (Celik et al., 2025) paper.

## F    MORE EXPERIMENT RESULTS

In this section, we show more tested experiments.

### F.1    ADDITIONAL FROM-SCRATCH RESULTS

**PPO**    As shown in Tab. 3, we use stabler parameter (num_minibatch=1), making PPO's data efficiency a little lower. We report PPO's training curve over a larger number of steps in Fig. 12. The results show that the final performance of our PPO implementation is comparable to or exceeds other open sources results.

**SAC**    To ensure a fair and rigorous comparison in our main results, we intentionally applied a unified set of hyperparameters for each method across all tasks. Consequently, SAC to exhibit poor performance on Ant-v4 under this unified settings. In this section, we conduct additional experiments under task-specific hyper-parameters in Tab. 8. Fig. 13 shows that with task-specific hyperparameters, SAC baseline can indeed converge to  4700 return on Ant-v4.

**Results on Tasks with Sparse Rewards**    We finally test on Robomimic-Can and OGBench-cube. Fig. 14 shows that all methods struggle on these two hard-exploration, sparse-reward tasks without pretrain, highlighting the necessity of offline-to-online training.

**Additional evaluation**    Fig. 15 shows the interquartile mean (IQM) with a 95% stratified bootstrap confidence interval as suggested by Agarwal et al. (2021).

We also report the probability-of-improvement Agarwal et al. (2021) in Fig. 16. Specifically, the probability of improvement metric estimates the likelihood that our algorithm $X$ outperforms a baseline $Y$ on a randomly selected task, formalized as $P(X > Y)$. Consistent with the method-

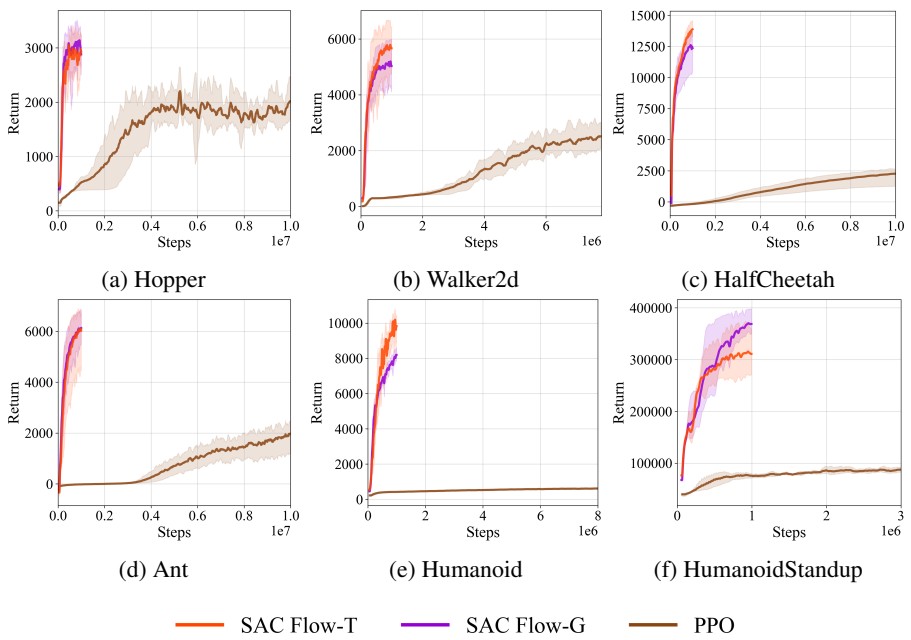

Figure 12: Training curve of PPO over a larger number of steps.

Table 8: Task-specific Hyperparameters for SAC Ant-v4

| Parameter | Value |
|---|---|
| Optimizer | Adam |
| Batch size ($M$) | 1024 |
| Replay buffer size | $1 \times 10^6$ |
| Discount factor ($\gamma$) | 0.99 |
| Policy learning rate | $2 \times 10^{-4}$ |
| Critic learning rate | $5 \times 10^{-4}$ |
| Target network update rate ($\tau$) | 0.001 |
| Learning starts | 50,000 |
| Entropy coefficient ($\alpha$) | 0.2 (initial value) |
| Target entropy | $-dim(\mathcal{A})$ |
| Automatic entropy tuning | True |
| Number of online environment steps | $1 \times 10^6$ |

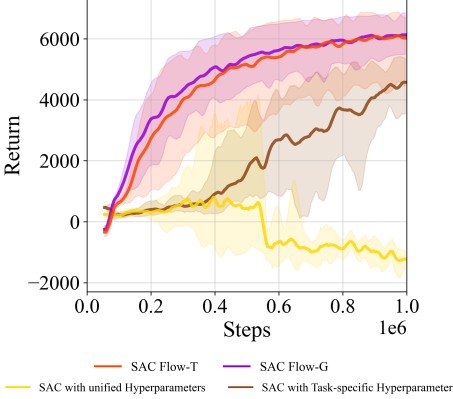

Figure 13: Training curves of SAC in Ant-v4. For the Ant task, SAC requires careful hyperparameter tuning to achieve stable convergence.

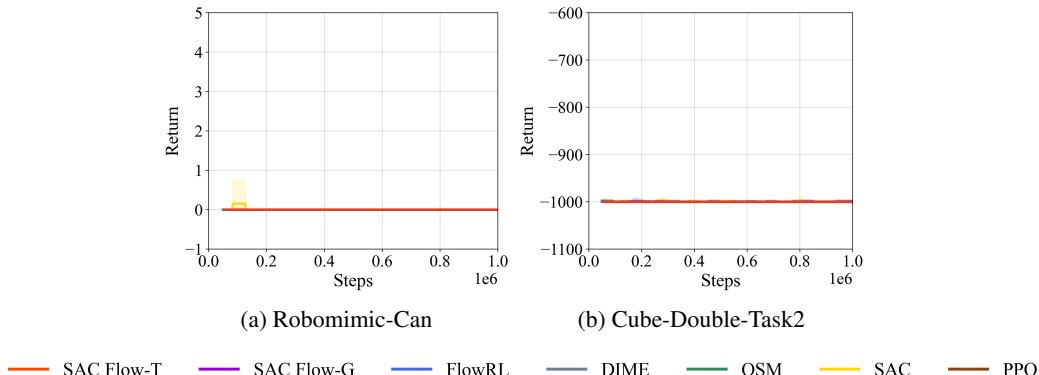

Figure 14: From-scratch training results on Robomimic-Can and OGBench-cube. All methods struggle on the hard-exploration, sparse-reward tasks without pretrain, highlighting the necessity of offline-to-online training.

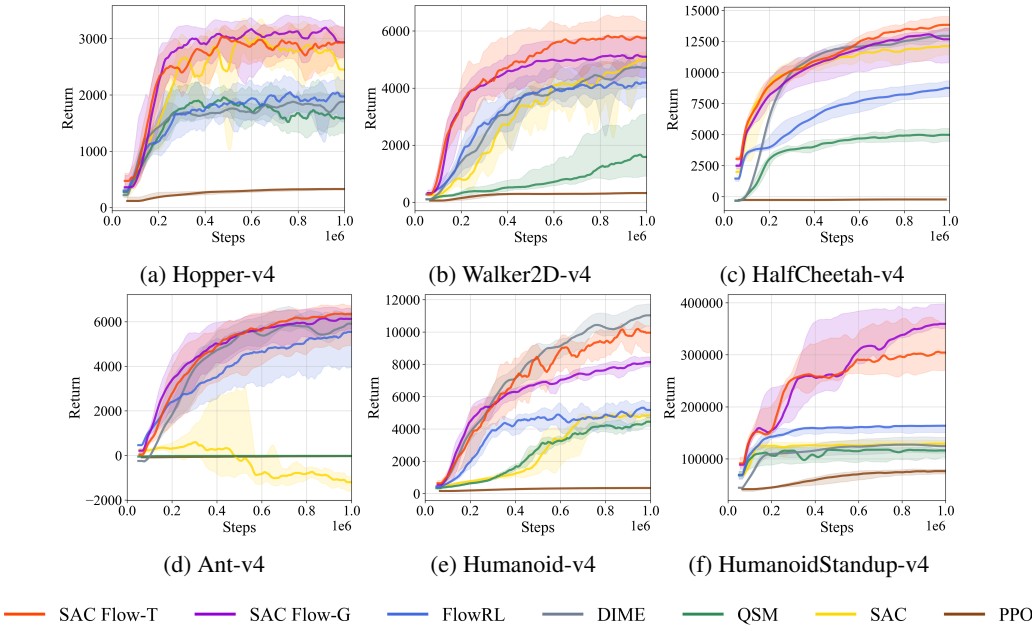

Figure 15: From-scratch training results on Mujoco (IQM return).

ology of Agarwal et al. (2021), we compute point estimates and 95% confidence intervals using stratified bootstrapping with 200 resamples across all tasks and seeds. This approach provides a robust pairwise comparison that mitigates the skewing effects of outlier performances often observed in aggregate metrics.

## F.2 ADDITIONAL OFFLINE-TO-ONLINE RESULTS

Fig. 17 abd Fig. 18 shows complete offline-to-online training performance in OGBench and Robomimic.

**Additional evaluation** Fig. 19 shows the interquartile mean (IQM) with a 95% stratified bootstrap confidence interval as suggested by Agarwal et al. (2021). We also report the probability-of-improvement in Fig. 20.

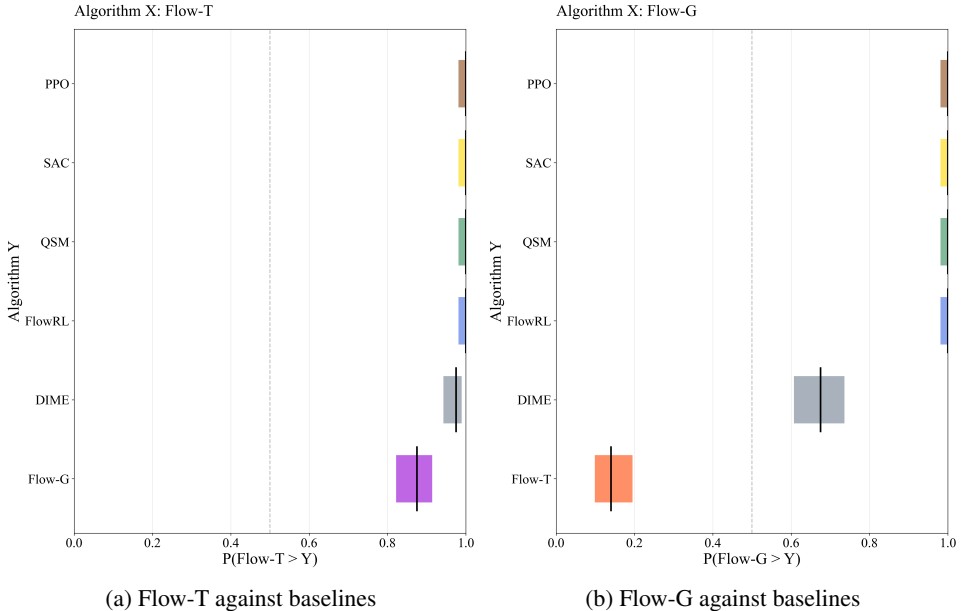

(a) Flow-T against baselines   (b) Flow-G against baselines

Figure 16: Probability-of-improvement in Mujoco benchmark. Each row shows the probability of improvement, with 95% bootstrap CIs, that the algorithm X on the top outperforms algorithm Y on the left, given that X was claimed to be better than Y.

### F.3 Additional Ablation Study

We further analyze the sensitivity to the specifics of the GRU and transformer. In main results, we set the default transformer parameter of Flow-T as layer=2, head=4, d_model=96. Fig. 21 shows our SAC Flow-T is robust to these three specifics of transformer.

For Flow-G, we analyze its sensitivity to gate network's width. We set the default gate width to 512. Fig. 22 shows that SAC Flow-G maintains stable convergence with gate width = 256. However, insufficient capacity in the gate network leads to performance degradation (gate width=64).

### F.4 Flow-T/G can be used in other off-policy RL algorithms

SAC flow stabilizes gradients during BPTT to ensure stable off-policy RL training, functioning independently of specific algorithms. We further extended our evaluation to TD3 Fujimoto et al. (2018). Fig. 23 demonstrates that our Flow-T/G architecture remains effective with TD3. In contrast, directly fine-tuning the flow using TD3 results in failure to converge or sub-optimal performance.

### F.5 Performance Analysis: Decoupling Architecture from Algorithm

To further investigate the source of our performance gains and explicitly disentangle the contribution of our architectural design (Flow-T) from the algorithmic objective, we conducted an additional ablation study comparing our method against DIME (Celik et al., 2025) under identical architectural conditions.

**DIME Flow-T Implementation.** Standard DIME optimizes a variational lower bound involving a forward process prior and a reverse path loss. To test whether these specific algorithmic components are the drivers of performance, we implemented **DIME Flow-T**. This variant retains the full DIME objective functions—including the injection of the prior gradient and the reverse path likelihood ratio—but replaces the standard backbone with our proposed Flow-T velocity parameterization.

**Results and Discussion.** We compared SAC Flow-T (our proposed method) against the new DIME Flow-T variant. The results are illustrated in Figure 24. We observe two key findings:

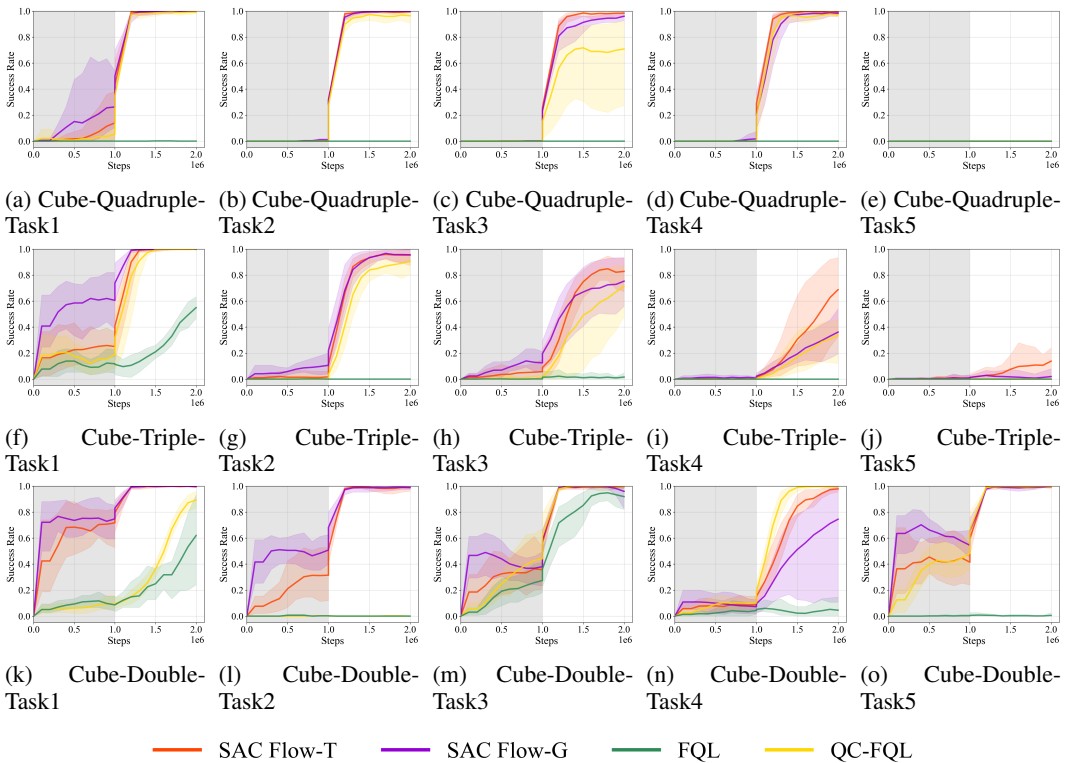

Figure 17: Complete offline-to-online training performance in OGBench. This figure illustrates the comprehensive training performance across all tasks. All methods are trained on 1M offline updates followed by 1M online interaction steps. Our methods, SAC Flow-T and SAC Flow-G, achieve competitive—often superior—performance across the evaluated benchmarks.

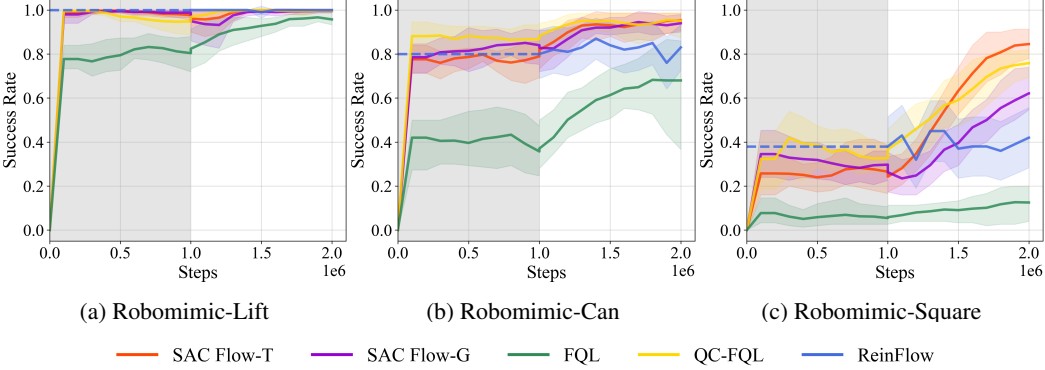

Figure 18: Complete offline-to-online training performance in Robomimic.

- Architecture enhances DIME: DIME Flow-T significantly outperforms the original DIME baseline (as shown in main results). This confirms that our Flow-T architecture provides substantial benefits in gradient stability and expressivity, regardless of the underlying RL objective.

- SAC Simplicity prevails: Crucially, as shown in Figure 24, **SAC Flow-T** consistently matches or outperforms DIME Flow-T (e.g., in HumanoidStandup). This suggests that the additional algorithmic complexity of DIME does not yield marginal gains once the velocity network is properly stabilized by Flow-T.

These results, combined with the successful application of Flow-T to TD3 (Section 5), demonstrate that the primary bottleneck in off-policy flow training is gradient instability in the rollout, which our

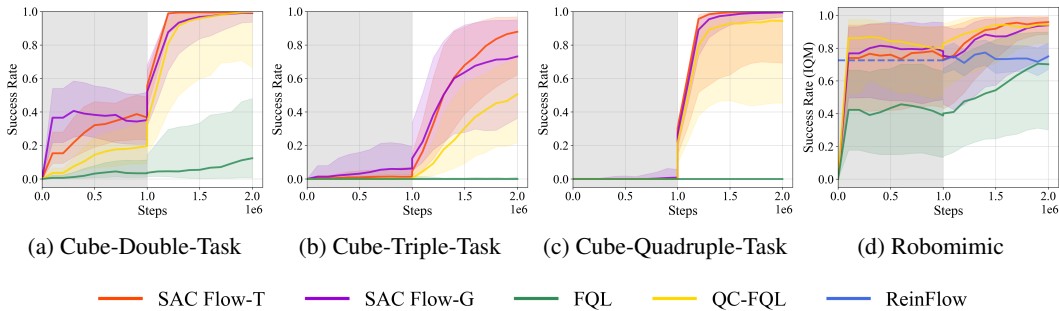

Figure 19: Aggregated offline-to-online performance on OGBench and Robomimic benchmarks (IQM).

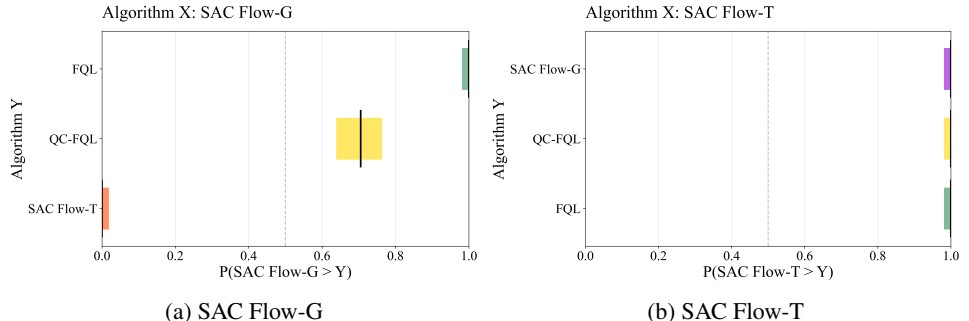

(a) SAC Flow-G                    (b) SAC Flow-T

Figure 20: Probability-of-improvement for offline-to-online setting on OGBench benchmarks.

Flow-T architecture effectively resolves. Once stabilized, the standard SAC objective is sufficient to achieve state-of-the-art performance.

# G    LLM USAGE DISCLOSURE

We used a large language model solely for writing polish. Its assistance was limited to grammar and style edits, wording suggestions for titles/abstract/captions, consistency of terminology, and minor LaTeX phrasing (e.g., figure/table captions and cross-reference text).

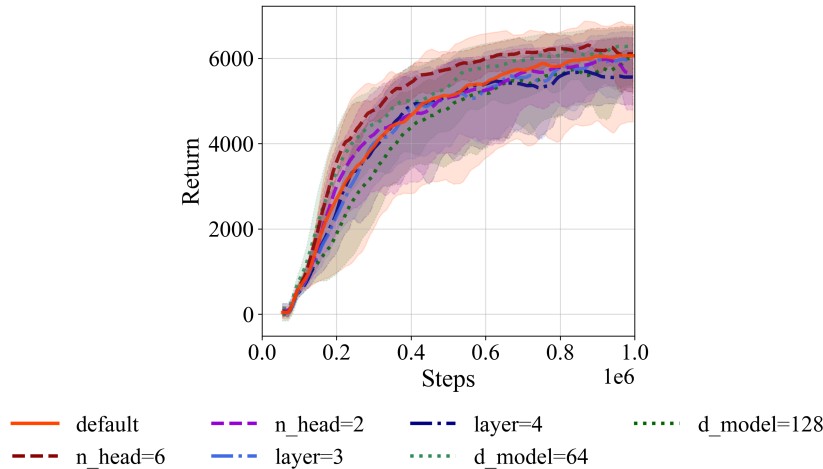

Figure 21: Ablation study on specifics of SAC Flow-T. Our SAC Flow-T are robust to the specifics of the transformers.

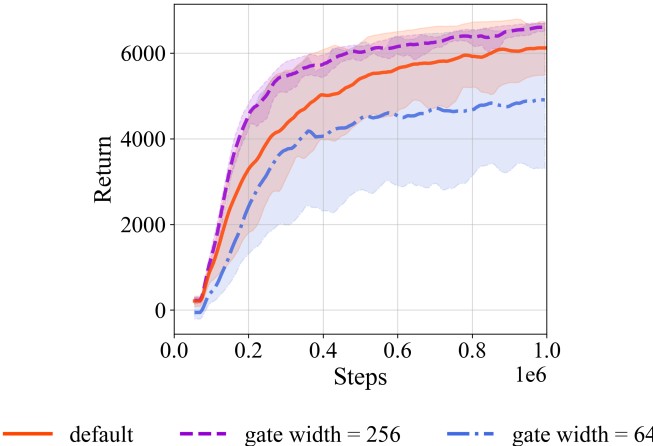

Figure 22: Ablation study on specifics of SAC Flow-G. SAC Flow-G maintains stable convergence with reduced gate widths. However, there exists degradation in final performance when the gate width is extremely small (64).

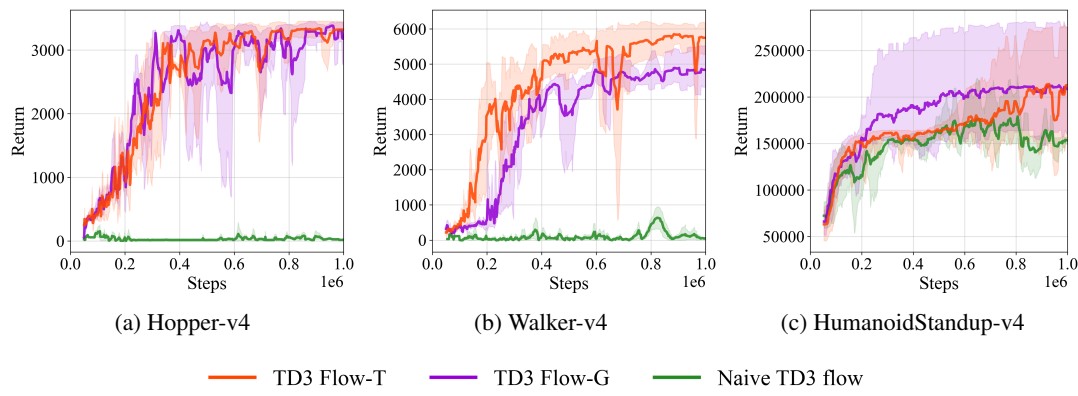

Figure 23: Evaluate our Flow-T/G architecture on TD3.

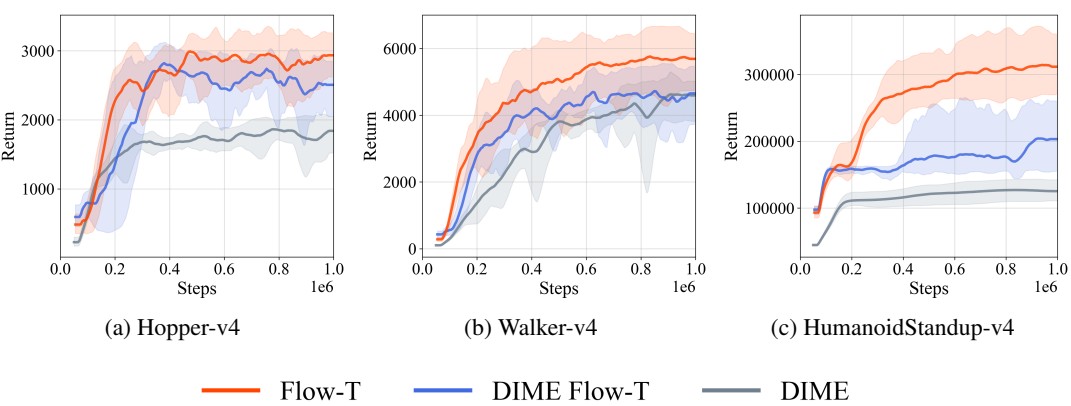

(a) Hopper-v4       (b) Walker-v4      (c) HumanoidStandup-v4

Flow-T     DIME Flow-T     DIME

Figure 24: Evaluate our Flow-T architecture on DIME loss.

