# OpenReview forum: "SAC Flow: Sample-Efficient Reinforcement Learning of Flow-Based Policies via Velocity-Reparameterized Sequential Modeling"
_ICLR.cc/2026/Conference — ICLR 2026 Poster_

### Official Review · Reviewer_dSSA · 2025-10-31

**Soundness:** 4
**Presentation:** 4
**Contribution:** 4
**Rating:** 8
**Confidence:** 3

**Summary:**

This paper proposes to learn flow policies using RL. Unlike previous work in behaviour cloning that learns flow fields that match the demonstration trajectories, this paper allows the flow field to be optimised against an explicit objective function. Previous work in RL for flow fields has experienced problems with gradient instability --- this paper shows that using a GRU or transformer to stabilise the gradients, and show learning improvement relative to baselines on a series of benchmarks.

**Strengths:**

Overall this paper proposes an interesting and novel modification to learning control flow fields. The paper is clearly written and the paper is careful to provide each step of SAC for flow fields, including the the proposed modifications. The authors show performance on both from-scratch problems and offline-to-online performance.

The experimental results show that the technique performs well on several baselines.

**Weaknesses:**

There are not significant weaknessess to the paper.

However, the gradient stabilisation that the authors propose, whether it is Flow-G or Flow-T, is (I think) essentially a form of regularisation. In eliminating the high variance motion of the gradient, the gradient is biased, and this may harm the learning. And we see that while both Flow-T and Flow-G generally outperform the baselines (except for Robomimic-Can and Cube-Double-Task2, where no techniques succeed), neither one is a clear winner across all benchmarks. This absence of a clear winner shows that the specific bias of the GRU or the Transformer is in fact limiting performance in some cases. It would be helpful to have some insight into how to control this bias, and whether or not the bias could be controlled, for instance by slowly widening the reset gate of the GRU or the cross-attention of the transformer.

I was also surprised that the GRU gradient norms are so much smaller for Flow-G and than for Flow-T (Figure 6). It would be good to understand how sensitive the performance is to the specifics of the GRU and transformer.

Given that this is a fairly modest step above flow-based RL, it would be good to have some understanding of the limitations of the technique. It is not clear if there are constraints on the reward function. Are there kinds of problems where this technique will not succeed?

**Questions:**

How sensitive the performance is to the specifics of the GRU and transformers?
Are there kinds of problems where this technique will not succeed?

---

> ### Author Response · Authors · 2025-11-18
> **Response 1**
>
> We sincerely thank the reviewer for their strong support and encouraging assessment of our work! We are delighted that you recognized the novelty of stabilizing flow-based policies via sequential modeling (GRU/Transformer) and appreciated the clarity of our SAC derivation. We are also glad that the experimental results across both from-scratch and offline-to-online settings demonstrated the effectiveness of our proposed technique. We appreciate your time and positive evaluation. Here are our responses:
>
> > However, the gradient stabilisation that the authors propose, whether it is Flow-G or Flow-T, is (I think) essentially a form of regularisation. In eliminating the high variance motion of the gradient, the gradient is biased, and this may harm the learning. And we see that while both Flow-T and Flow-G generally outperform the baselines (except for Robomimic-Can and Cube-Double-Task2, where no techniques succeed), neither one is a clear winner across all benchmarks. This absence of a clear winner shows that the specific bias of the GRU or the Transformer is in fact limiting performance in some cases. It would be helpful to have some insight into how to control this bias, and whether or not the bias could be controlled, for instance by slowly widening the reset gate of the GRU or the cross-attention of the transformer.
>
>
> Thank you for your insight. The biased gradient you mentioned is indeed a potential concern. We believe two primary sources of bias exist in the problem studied in this paper.
>
> First, regarding the optimization objective: as detailed in Section 4, we utilize the standard SAC loss, which ensures our training objective is unbiased.
>
> Second, the technique of GRU/Transformer-decoder velocity unavoidably introduces the inductive bias inherent to these architectures [1][2]. This architectural bias may cause slight variations in the policy's convergence behavior. We identify two potential methods to control this:
>
> - As you suggested, we could mitigate this bias by slowly widening the GRU's reset gate, adjusting the Transformer's cross-attention, or employing different attention masks.
>
> - It's possible to utilize specific initialization techniques. For instance, initializing the GRU gate cells to 1 or 0 might yield different convergence behaviors.
>
> Your suggestion highlights an interesting direction for future work. We plan to quantitatively evaluate and further investigate methods to mitigate the impacts of such architectural variations.
>
>
> ---
>
>
> > I was also surprised that the GRU gradient norms are so much smaller for Flow-G and than for Flow-T (Figure 6). It would be good to understand how sensitive the performance is to the specifics of the GRU and transformer.
>
>
> We thank the reviewer for raising the constructive question. We have included additional experiments in Appendix F.3 to analyze how performance varies with respect to the GRU and Transformer specifics.
>
> Specifically, for Flow-T (which defaults to 2 layers, 4 heads, and d\_model=96), we evaluated the sensitivity to these three parameters in the Ant-v4 environment. The results in Fig. 21 indicates that our SAC Flow-T are robust to the specifics
> of the transformers.
>
> For Flow-G (which defaults to gate width = 512), we change the gate width to 256 and 64 to analyze its sensitivity. As shown in Fig. 22, SAC Flow-G maintains stable convergence even with reduced gate widths. However, we observe some degradation in final performance when the gate width is extremely small (e.g., 64).
>
> We hope these supplementary experiments prove helpful.
>
> ---

---

> ### Author Response · Authors · 2025-11-18
> **Response 2**
>
> > Given that this is a fairly modest step above flow-based RL, it would be good to have some understanding of the limitations of the technique. It is not clear if there are constraints on the reward function. Are there kinds of problems where this technique will not succeed?
>
>
> Thank you for your question.
>
> Our proposed SAC Flow is not highly constrained by the reward function. As demonstrated in Table 1, it consistently achieves high performance on benchmarks with both dense rewards (e.g., Mujoco) and sparse rewards (e.g., OGBench & Robomimic).
>
> Regarding limitations, while this work successfully resolves the gradient instability issue for flow policies during off-policy training, several limitations remain.
>
> - First, the velocity re-parameterization technique in SAC Flow imposes structural constraints on the actor (as illustrated in Figure 3). This makes it difficult to apply SAC Flow to tasks that have pre-defined actor architectures. A key example is training a flow-based action expert for a Vision-Language-Action (VLA) model, such as Pi0 [3] or Pi0.5 [4]. These VLA models typically utilize a pre-defined flow policy structure that has undergone extensive pre-training, making it challenging to integrate our velocity re-parameterization.
>
> - Furthermore, SAC Flow's improvements are focused on the actor, and still employs a standard SAC critic. A future research direction is to combine SAC Flow with more advanced critic techniques to further unlock the full potential of flow-based models.
>
> ---
>
> **References:**
>
> [1] Ishii, Taiga, Ryo Ueda, and Yusuke Miyao. "Empirical analysis of the inductive bias of recurrent neural networks by discrete Fourier transform of output sequences." arXiv preprint arXiv:2305.09178 (2023).
>
> [2] Chung, Junyoung, et al. "Empirical evaluation of gated recurrent neural networks on sequence modeling." arXiv preprint arXiv:1412.3555 (2014).
>
> [3] Black, Kevin, et al. "$\pi_0 $: A Vision-Language-Action Flow Model for General Robot Control." arXiv preprint arXiv:2410.24164 (2024).
>
> [4] Intelligence, Physical, et al. "π0. 5: a vision-language-action model with open-world generalization, 2025." URL https://arxiv. org/abs/2504.16054 1.2: 3.

---

### Official Review · Reviewer_bZhq · 2025-11-01

**Soundness:** 2
**Presentation:** 3
**Contribution:** 2
**Rating:** 4
**Confidence:** 3

**Summary:**

This paper introduces **SAC Flow**, an off-policy reinforcement learning algorithm for **flow-based policies**.
It reinterprets the multi-step flow rollout as a **residual RNN**, linking instability in prior methods to recurrent gradient issues.
To stabilise training, the authors reparameterise the **velocity network** using sequential architectures: **Flow-G** (GRU-gated) and **Flow-T** (Transformer-decoded).
A **noise-augmented rollout** allows tractable entropy computation in SAC.
Experiments on **MuJoCo**, **OGBench**, and **Robomimic** report state-of-the-art sample efficiency over existing diffusion- and flow-based baselines, with ablations assessing gradient stability and robustness.

**Strengths:**

* **Novel conceptual framing:** The sequential-model reinterpretation of flow rollouts (residual RNN ≈ flow ODE) is insightful and could influence future flow-based RL research.
* **Architectural innovation:** Flow-G and Flow-T represent principled design choices inspired by GRUs and Transformers, addressing stability through gating and pre-norm residuals.
* **Algorithmic completeness:** The integration into SAC with a noise-augmented rollout is well-motivated and yields a self-contained training procedure.
* **Empirical breadth:** Evaluations cover diverse benchmarks (MuJoCo, OGBench, Robomimic) and both from-scratch and offline-to-online regimes.
* **Reproducibility:** Appendices provide detailed architectures and training algorithms.

**Weaknesses:**

**Summary:**
The paper contains a **promising conceptual contribution** — interpreting flow-based policies as sequential models and borrowing stability mechanisms from RNN/Transformer design — and achieves competitive empirical performance.
However, the **central motivation (gradient instability in naïve SAC-Flow)** is **not convincingly evidenced**, and several key figures are inconsistent or uninformative.
The experimental section needs stronger causal and statistical support to meet ICLR’s bar for empirical soundness.

### **Weaknesses**

#### **1. Insufficient evidence for the claimed instability**

The paper’s central claim — that “Directly backpropagating through the K-step action sampling is often unstable” — lacks quantitative support.
Even the ablation on velocity network parameterizations (Fig. 13a) shows gradient explosion only near the end of training, while Fig. 6(a) implies immediate collapse if I'm not mistaken.
These two plots contradict each other, and neither connects gradient norms to learning-curve failure.
Without aligned gradient-and-performance analysis, the claim of severe instability remains unproven.
So, I felt that aggregated plot of Fig. 6(a) is therefore potentially misleading and should be clarified or replaced with joint plots of gradient magnitude and return.

#### **2. Ambiguous or uninformative results (Figs. 4g,h)**

Panels (g) *Robomimic-Can* and (h) *Cube-Double-Task2* show all methods failing equally.
The authors acknowledge this only briefly as motivation for the offline-to-online setup (p. 7), but these subfigures add no evaluative value.
They should either be removed or replaced with quantitative evidence (e.g., normalized success rates, exploration metrics) illustrating why SAC Flow fails on sparse rewards.

#### **3. Lack of aggregate evaluation across environments**

Per-task curves make it difficult to judge overall performance.
Following **Agarwal et al., NeurIPS 2021**, the authors should report **aggregate IQM returns** with **95 % bootstrap confidence intervals** and **probability-of-improvement** or **rank-based summaries** across all benchmarks.
This would clarify whether SAC Flow-T and -G consistently outperform baselines rather than winning selectively.
* Agarwal et al., NeurIPS 2021 — “Deep Reinforcement Learning at the Edge of the Statistical Precipice.”

#### **4. Missing causal evidence for performance gains**

The connection between gradient stabilization (Flow-G/T) and improved returns is asserted but not verified through controlled experiments isolating that factor.
So, it's related to my first point but without a quantitative baseline showing naïve SAC-Flow failure, it is unclear whether the gains stem from the proposed architectures or from implementation details.

**Questions:**

see weakness

---

> ### Author Response · Authors · 2025-11-18
> **Response 1**
>
> We appreciate the reviewers' constructive comments. Our revisions and detailed responses are provided below.
>
> > **1. Insufficient evidence for the claimed instability.** The paper’s central claim — that “Directly backpropagating through the K-step action sampling is often unstable” — lacks quantitative support. Even the ablation on velocity network parameterizations (Fig. 13a) shows gradient explosion only near the end of training, while Fig. 6(a) implies immediate collapse if I'm not mistaken. These two plots contradict each other, and neither connects gradient norms to learning-curve failure. Without aligned gradient-and-performance analysis, the claim of severe instability remains unproven. So, I felt that aggregated plot of Fig. 6(a) is therefore potentially misleading and should be clarified or replaced with joint plots of gradient magnitude and return.
>
> ---
>
> We thank the reviewer for the careful examination of our experimental results. We appreciate the opportunity to clarify the connection between gradient behavior and learning performance, and we address the specific concerns as follows:
>
> **(1) Clarification on Figure Correspondence**
> We respectfully suggest that there may be a misunderstanding regarding the mapping between the learning curves in the main paper and the gradient plots in the Appendix. We apologize that our original appendix captions did not make this correspondence sufficiently clear.
> Specifically, **Figure 13a** in the original submission illustrates the gradient norms for **HalfCheetah**, whereas the learning curve in **Figure 6(a)** corresponds to **Walker2d**. The gradient plot that actually corresponds to Walker2d is **Figure 12a**, which indeed shows significant instability early in training, aligning perfectly with the performance collapse observed in Figure 6(a). Thus, the results are consistent and do not contradict each other.
>
> **(2) New Aligned Gradient-and-Performance Analysis**
> We fully agree with the reviewer’s constructive suggestion that plotting gradient norms in isolation is insufficient to conclusively demonstrate causality. To address this, **we have replaced Figure 6 in the main paper with a new, aligned analysis** (see Section 5.3 and revised Fig. 6).
> This updated figure now presents the **Episodic Return** and **Gradient Norms** side-by-side for the same training run. The new results clearly demonstrate that the learning failure of the Naive SAC Flow baseline is directly synchronized with extreme gradient instability (exploding/oscillating norms). In contrast, our proposed **SAC Flow-T** and **SAC Flow-G** maintain well-conditioned gradient norms throughout the entire training process, enabling stable convergence. We believe this joint analysis provides robust quantitative evidence for our central claim regarding the instability of direct backpropagation in standard flow policies. Meanwhile, for the aligned analysis is shown, we have removed Figure 12 and Figure 13 in the Appendix.
>
> > **2. Ambiguous or uninformative results (Figs. 4g,h)** Panels (g) Robomimic-Can and (h) Cube-Double-Task2 show all methods failing equally. The authors acknowledge this only briefly as motivation for the offline-to-online setup (p. 7), but these subfigures add no evaluative value. They should either be remwozoved or replaced with quantitative evidence (e.g., normalized success rates, exploration metrics) illustrating why SAC Flow fails on sparse rewards.
>
>
> We appreciate the reviewer pointing out the issue with the arrangement of our plots. We have moved panels (g) and (h) from Fig. 4 to Appendix F.
>
> We also wish to clarify that SAC does not fail on sparse rewards, but rather fails on sparse rewards when **trained from scratch**. The results (,Fig. x originally panels (g) and (h) in Fig. 4) were intended to illustrate this point. In the offline-to-online setting, SAC-based methods are also capable of solving sparse reward problems, as demonstrated in Fig. 5.
>
> ---

---

> ### Author Response · Authors · 2025-11-18
> **Response 2**
>
> > **3. Lack of aggregate evaluation across environments** Per-task curves make it difficult to judge overall performance. Following Agarwal et al., NeurIPS 2021, the authors should report aggregate IQM returns with 95\% bootstrap confidence intervals and probability-of-improvement or rank-based summaries across all benchmarks. This would clarify whether SAC Flow-T and -G consistently outperform baselines rather than winning selectively.
>
> We thank the reviewer for this valuable recommendation. We agree that aggregate metrics based on stratified bootstrapping provide a more robust and unbiased assessment of overall performance than individual learning curves.
>
> In response, we have conducted a comprehensive aggregate evaluation following the protocols (IQM and probability-of-improvement) of Agarwal et al. (2021). These results are now detailed in Appendix F.1 and F.2 (specifically Figures 15, 16, 19, and 20). The analysis yields the following conclusions:
>
> 1.  Interquartile Mean (IQM): As shown in the IQM plots (Figs. 15 & 19), our methods consistently achieve state-of-the-art performance with tight confidence intervals. This confirms that the superiority of our approach is robust across the benchmark suite and is not driven by outliers.
>
> 2.  Probability-of-improvement (POI): The POI analysis provides strong statistical support for the consistency of our performance gains.
>     * SAC Flow-T exhibits a POI of $\approx 1.0$ against all baselines except DIME, where it maintains a dominant probability of 95\%.
>     * SAC Flow-G similarly achieves a POI of $\approx 1.0$ against most baselines, with a probability of 70\% against DIME.
>
> These metrics statistically verify that SAC Flow-T and SAC Flow-G provide a significant and consistent improvement over existing methods, rather than winning selectively on specific tasks.
>
> ---
>
> > **4. Missing causal evidence for performance gains** The connection between gradient stabilization (Flow-G/T) and improved returns is asserted but not verified through controlled experiments isolating that factor. So, it's related to my first point but without a quantitative baseline showing naïve SAC-Flow failure, it is unclear whether the gains stem from the proposed architectures or from implementation details.
>
>
> As this concern is closely related to **Question 1**, we respectfully refer the reviewer to our detailed response above and the **newly added Figure 6**.
>
> To explicitly address the causality:
> 1.  Controlled Experiment: The "Naive SAC Flow" baseline presented in the revised Figure 6 serves as the controlled variable. It shares the exact same implementation details (loss functions, hyperparameters, code base) as our proposed methods, differing only in the velocity network architecture (standard Flow vs. our Flow-G/T). The results unequivocally show that the Naive baseline fails to learn due to gradient explosion, whereas our methods stabilize the gradients and succeed. This isolates our architectural reparameterization as the direct cause of the performance gain.
>
> 2.  Literature Context: We further note that the instability of directly optimizing flow-based policies via off-policy RL is a recognized "long-standing problem" in the field. Recent works, such as **FQL [1]** and **FlowRL [2]**, explicitly highlight the numerical instability of backpropagating through flow/ODE steps as a critical bottleneck, often resorting to distillation or surrogate objectives to bypass it. Our work directly addresses this known barrier, demonstrating that stability can be achieved architecturally.
>
> **References:**
>
> [1] Park, S., Li, Q., & Levine, S. (2025). Flow Q-learning.
>
> [2] Lv, L., Li, Y., Luo, Y., Sun, F., Kong, T., Xu, J., & Ma, X. (2025). Flow-Based Policy for Online Reinforcement Learning.

---

> > ### Comment · Reviewer_bZhq · 2025-11-24
> > **Response**
> >
> > Thank you for the effort you put into the revisions and for addressing the reviewers’ concerns.
> >
> > I appreciate the updated Figure 6 — the aligned gradients-and-performance plots do help clarify your argument. However, based on the revised figures for **Ant** and **Walker2d**, I still find the evidence somewhat mixed for your central claim that:
> >
> > > “Directly backpropagating through the K-step action sampling is often unstable.”
> >
> > In Ant, the correspondence between exploding gradients and performance collapse is clear.
> > But in Walker2d, the pattern is less convincing: the gradient norms peak mid-training rather than near failure, and the magnitude does not appear particularly large. This makes it harder to interpret the instability as the primary cause of the poor performance in that case.
> >
> > Aside from this point, my earlier concerns (2, 3, and 4) are addressed in the revision.

---

> ### Author Response · Authors · 2025-11-24
> **Response to Follow-up**
>
> We sincerely thank the reviewer for the continued engagement and for raising the insightful observation regarding the magnitude differences between Ant and Walker2d.
>
> We fully agree with the reviewer that observing gradient norms in isolation can be ambiguous because their absolute magnitudes are **task-dependent**. The magnitude of the policy gradient is intrinsically linked to the scale of the Q-function ($\nabla_a Q(s,a)$) and the specific reward structure of the environment.
>
> For example, although Ant-v4 naturally exhibits larger gradients due to its reward scale, a norm of ~50 in Walker2d, while numerically smaller, actually represents a massive 10x explosion relative to the "healthy" norm of ~5 observed in our stable baselines, constituting a deviation just as destructive to its specific optimization landscape as the larger absolute spikes seen in Ant.
>
> To rigorously quantify this "relative instability" and provide a unified explanation across tasks with varying scales, we have expanded our analysis in the **following ways**:
>
> *  **Added Hopper-v4:** We included a third environment to verify if the pattern holds across different dynamics.
> *  **Introduced "Gradient Ratio" Analysis:** Instead of looking at absolute raw values, we computed the **ratio** of gradient norms between the unstable Naive SAC Flow and the stable SAC Flow-G. This metric normalizes the task-specific scales (e.g., dividing the "unstable 50" by the "stable 5" in Walker2d), allowing us to isolate the instability factor from the environment's inherent scale.
>
> **Our Findings (visualized in the updated Figure 6, bottom row):**
> We observe a striking consistency across all three tasks (Ant, Walker2d, Hopper):
>
> * **The "Tenfold" Phenomenon:** Despite the varying absolute scales, the Gradient Norm Ratio (Naive / Flow-G) consistently spikes to **approximately tenfold (10x)** the stable magnitude in all environments.
> * **Correlation with Failure:** Crucially, this relative explosion typically peaks around **$10^5$** steps. Looking at the corresponding learning curves (Fig. 6 a, f, k), this is precisely the moment when the Naive SAC Flow stops learning and its performance begins to collapse or stagnate.
>
> **Conclusion:**
> This comparative analysis suggests that the "instability" is defined not by a specific absolute threshold (e.g., norm > 100), but by the relative explosion of the gradient compared to the stable baseline. The Naive rollout consistently produces gradients that are an order of magnitude noisier/larger than what the optimization landscape requires, leading to the observed failure in all three tasks.
>
> We believe this unified view, supported by the new Hopper experiment and the Ratio plots, conclusively connects gradient instability to training failure. **We hope that this new evidence and the unified analysis fully address your remaining concerns. In light of these clarifications and the strengthened empirical support for our central claim, we respectfully ask that you reconsider your evaluation of our paper.**

---

> ### Author Response · Authors · 2025-11-27
> **Inquiry regarding revision**
>
> Dear Reviewer bZhq,
>
> We sincerely thank you for your constructive feedback, which has greatly helped us improve the quality of our manuscript.
>
> As the discussion phase deadline is approaching, we are writing to check if our submitted revision and responses have adequately addressed your previous concerns. We are more than happy to answer any remaining questions or provide further clarifications if needed.
>
> We look forward to hearing from you.
>
> Best regards,
>
> The Authors

---

> > ### Comment · Reviewer_bZhq · 2025-11-28
> > **Response**
> >
> > Thank you for the additional follow-up. My concerns have been resolved, so I have raised my score to 6.

---

### Official Review · Reviewer_rXyd · 2025-11-01

**Soundness:** 3
**Presentation:** 3
**Contribution:** 4
**Rating:** 6
**Confidence:** 4

**Summary:**

The paper tackles instability when training flow-based policies with off-policy RL by viewing the K-step rollout as a deep residual RNN and proposing two stabilizing architectures that resemble a GRU-style gated cell and a Transformer decoder.

**Strengths:**

- The problem is meaningful and of great importance to flow policies applied to RL
- The proposed approach provides new perspectives to solve the problem, and the resulting solution is clean and simple
- Empirical results demonstrate the strong performance of their approach, mostly matching or exceeding the state of the art

**Weaknesses:**

- Line 159 “Off-policy methods (e.g., SAC, TD3) are highly sample-efficient (Mambelli et al., 2024), yet directly backpropagating through the K-step action sampling is often unstable, especially for large K (Park et al., 2025).” — I think this core technical challenge deserves a more serious presentation with necessary math beyond verbal explanations.
- The current presentation of Flow-G/T assumes that the architectures are natural choices inherent from modern improvement over RNNs. However, more mathematical details, e.g., showing the Jacobians after the proposed fixes, might strengthen the authors' argument
- Figure 4: I am a bit confused about the results presented in Figure 4. For example, why is the reward for PPO so low, and looks like it just barely learns anything? The results do not match (are not even close to) common open source implementations, such as the ones reported in CleanRL (https://docs.cleanrl.dev/rl-algorithms/ppo/#experiment-results_2). Also, SAC does not seem to solve Ant-v4. That said, can the authors report how they came to the chosen hyperparameter configuration for all the algorithms?
- Minor:
    - “State-of-the-art sample efficiency and performance.” — This is not counted as a separate contribution, but more of the evaluation results of the first two contributions
    - Color scheme chosen in a number of figures, e.g., Figs. 4 and 7, is visually indistinguishable

**Questions:**

- Line 132 “yet a single unimodal Gaussian cannot capture inherently multimodal action distributions, a limitation that is especially harmful in long-horizon robotic control” —  Actions being unimodal and the horizon of the tasks seem to be two rather independent things, can the authors elaborate more on this point?
- Line 398 “Among all experiments, the sampling steps of flow-based policies are set to 4, and the denoising steps of diffusion policies are set to 16. More details of the experimental setting are described in Appendix C and Appendix D.” — How are the 4 steps and 16 steps determined respectively

---

> ### Author Response · Authors · 2025-11-18
> **Response 1**
>
> We sincerely thank the reviewer for their constructive feedback and the effort dedicated to improving the quality of this work. We provide our detailed, point-by-point responses below.
>
> > Line 159 “Off-policy methods (e.g., SAC, TD3) are highly sample-efficient (Mambelli et al., 2024), yet directly backpropagating through the K-step action sampling is often unstable, especially for large K (Park et al., 2025).” — I think this core technical challenge deserves a more serious presentation with necessary math beyond verbal explanations.
>
> We thank the reviewer for this valuable feedback. We agree completely that this core technical challenge is the central motivation for our work and deserves a formal mathematical presentation.
>
> In our original submission, we kept this discussion high-level in the main text to maintain the paper's flow. Based on the reviewer's suggestion, we have added a new, detailed mathematical analysis in Appendix B.1 (which we briefly reference in the end of Section 3).
>
> In this new appendix, we now formally derive the exploding/vanishing gradient problem. We show that the backpropagation of the actor's loss $L$ through the $K$-step rollout (viewed as an RNN, per our Eq. (6)) results in a long product of Jacobian matrices:
> $$
> \nabla_{A_{t_0}} L = (\nabla_{A_{t_K}} L) \cdot \prod_{i=0}^{K-1} \left( I + \frac{\partial f_\theta(t_i, A_{t_i}, s)}{\partial A_{t_i}} \right)
> $$
> Following the seminal analysis of BPTT [1], we explain why this long product is exponentially unstable, leading to the gradient pathologies that motivate our work. We acknowledge that BPTT instability is a well-studied phenomenon in the sequential modeling literature. To ground our analysis in this established theory, we have added citations [1-3] to the main text, referring interested readers to these works for a comprehensive treatment of the underlying gradient dynamics.
>
>
> ---
>
>
>
> > The current presentation of Flow-G/T assumes that the architectures are natural choices inherent from modern improvement over RNNs. However, more mathematical details, e.g., showing the Jacobians after the proposed fixes, might strengthen the authors' argument
>
> This is an excellent point, which is highly related to the first comment. We agree that a mathematical justification for why Flow-G and Flow-T are principled solutions (and not just "natural choices") is crucial.
>
> We have addressed this in the second part of our new Appendix B.2. After establishing why the standard RNN formulation is unstable (as described in our response above), we now provide a detailed mathematical motivation for our proposed architectures:
>
> 1.  For Flow-G: We show how its GRU-style gated update, $A_{t_{i+1}} = (I - \Delta t_i g_i) A_{t_i} + \dots$, allows the network to learn to set its step-wise Jacobian $\frac{\partial A_{t_{i+1}}}{\partial A_{t_i}} \approx I$ (by driving the gate $g_i \to 0$). This dynamically creates a stable gradient path.
>
> 2.  For Flow-T: We explain that its stability arises from the combination of Pre-Layer Normalization and Residual Connections. We discuss how this specific pairing (now standard in Transformers) was explicitly designed to stabilize deep gradient propagation by bounding the Jacobians of the sub-layers, thus keeping the overall Jacobian product $\nabla_{A_{t_0}} L$ well-conditioned.
>
> We also explicitly state that while it is intractable to provide a strict mathematical proof that the full Jacobian product is always 1 (unlike the simple case in LSTMs, due to the complexity of modern networks), our architectures import the precise mechanisms that were engineered to solve this exact problem, providing a strong inductive bias that is empirically validated by our ablation studies (Fig. 6).
>
> We believe this new appendix section now provides the requested "mathematical details" and strongly reinforces our argument that Flow-G and Flow-T are principled and effective solutions to the core instability problem.
>
>
> ---

---

> ### Author Response · Authors · 2025-11-18
> **Response 2**
>
> > Figure 4: I am a bit confused about the results presented in Figure 4. For example, why is the reward for PPO so low, and looks like it just barely learns anything? The results do not match (are not even close to) common open source implementations, such as the ones reported in CleanRL (https://docs.cleanrl.dev/rl-algorithms/ppo/#experiment-results_2). Also, SAC does not seem to solve Ant-v4. That said, can the authors report how they came to the chosen hyperparameter configuration for all the algorithms?
>
>
>
> We thank the reviewer for pointing out this issue. In response, we have detailed the hyperparameters in Appendix E.1.
>
> Regarding the PPO results, we opted for a conservative num_minibatch=1  (whereas CleanRL uses 32) following PPO based flow/diffusion methods [7][8]. This setting ensures stable training, particularly in the challenging humanoid-standup task. This choice resulted in comparatively lower data efficiency, which explains the limited growth in returns within the initial 1M steps. To provide a clearer picture, we now report PPO's training curve over a larger number of steps in Appendix F.1. These results show that the final performance of our PPO implementation is comparable to or exceeds the results reported by CleanRL (https://docs.cleanrl.dev/rl-algorithms/ppo/#experiment-results_2).
>
> Regarding SAC's performance on the Ant-v4, we observe that Ant's large exploration space necessitates more conservative hyperparameters. However, to ensure a fair and rigorous comparison in our main results, we intentionally applied a unified set of hyperparameters for each method across all tasks. Consequently, the SAC baseline struggled to converge on Ant-v4 under these **unified settings**. We conduct additional experiments to indicate that with **task-specific hyperparameters** (specifically, a smaller target update ratio $\tau$, a lower lr, and a larger batch size), SAC can indeed converge to ~4700 return on Ant-v4. We have included these supplementary results in Appendix F.1.
>
> We hope our supplementary experiments address your concerns.
>
> ---
>
> > Minor:
> >>“State-of-the-art sample efficiency and performance.” — This is not counted as a separate contribution, but more of the evaluation results of the first two contributions
>
> Thank you for highlighting this issue with our representation. We have revised "Contribution 3" to **"Extensive experimental evaluation"** to better reflect that it serves as the evaluation of the first two contributions.
>
>
> >>Color scheme chosen in a number of figures, e.g., Figs. 4 and 7, is visually indistinguishable
>
> Thanks for pointing out that. We have change the color scheme in Fig. 4 and linestyle in Fig. 7 to make the plots more clear.
>
> ---

---

> ### Author Response · Authors · 2025-11-18
> **Response 3**
>
> > Line 132 “yet a single unimodal Gaussian cannot capture inherently multimodal action distributions, a limitation that is especially harmful in long-horizon robotic control” — Actions being unimodal and the horizon of the tasks seem to be two rather independent things, can the authors elaborate more on this point?
>
>
> We thank the reviewer for this insightful question, which highlights an important point we did not make sufficiently clear.
>
> The reviewer is correct that for a standard, single-step (Markovian) policy $p(a_t | s_t)$, the task horizon and the need for action modality are indeed largely independent.
>
> Our statement was implicitly referring to the common and effective practice of using **temporally extended actions (i.e., "action chunking")** to solve complex, long-horizon tasks. This is a key technique used in several modern robotics benchmarks and baselines, including those in our own offline-to-online experiments (e.g., OGBench and Robomimic, following methods like QC-FQL[4]).
>
> When a policy is trained to output an action sequence or "chunk" $a_{t:t+H}$ (where $H > 1$), the distribution over this high-dimensional action chunk space, $p(a_{t:t+H} | s_t)$, is often inherently multimodal. For example, from a given state, an agent might have two distinct, viable long-term strategies (e.g., a 2-second "grasp" sequence vs. a 2-second "push" sequence). A unimodal Gaussian policy would fail by averaging these distinct strategies into a single, non-functional "mean" action chunk.
>
> Therefore, the limitation of unimodal policies becomes especially harmful in these long-horizon settings precisely because they are incompatible with the multimodal action chunks required for effective, temporally-extended decision-making.
>
> We apologize for this lack of clarity in the original text. We have revised the manuscript in Section 2.3 to explicitly state this connection between long-horizon tasks, action chunking, and the need for multimodal policies.
>
>
> ---
>
>
>
> > Line 398 “Among all experiments, the sampling steps of flow-based policies are set to 4, and the denoising steps of diffusion policies are set to 16. More details of the experimental setting are described in Appendix C and Appendix D.” — How are the 4 steps and 16 steps determined respectively
>
> We thank the reviewer for requesting this clarification. Our choices for these hyperparameters were based on two primary factors:
>
> 1.  Fundamental Model Efficiency: The step difference (4 for flow vs. 16 for diffusion) reflects the models' inherent efficiencies. Flow-based models (particularly Rectified Flow) are highly efficient, requiring very few steps (e.g., 2-10) for high-quality generation. In contrast, diffusion models typically require significantly more denoising steps (e.g., 10-50) to achieve similar-quality results.
>
> 2.  Fair Baseline Comparison: To ensure our comparisons are fair and rigorous, we adopted the standard hyperparameter settings used in the respective state-of-the-art baseline papers.
>     * Our choice of 4 sampling steps for our flow-based policies (SAC Flow-G and SAC Flow-T) directly follows the setting used in the key flow-based baseline, ReinFlow [5].
>     * Our choice of 16 denoising steps for the diffusion policy baselines (e.g., DIME) follows the setting used in the DIME [6] paper.
>
> We have added a new paragraph, "Sampling Steps Justification", to Appendix E (Implementation Details) to make this reasoning explicit in the revised manuscript.
>
> ---
>
> **References:**
>
> [1] Hochreiter, S., & Schmidhuber, J. (1997). Long short-term memory. Neural computation.
>
> [2] Pascanu, R., Mikolov, T., & Bengio, Y. (2013). On the difficulty of training recurrent neural networks. In International conference on machine learning .
>
> [3] Bengio, Y., Simard, P., & Frasconi, P. (1994). Learning long-term dependencies with gradient descent is difficult. IEEE transactions on neural networks.
>
> [4] Li, Q., Zhou, Z., & Levine, S. (2025). Reinforcement learning with action chunking. arXiv preprint arXiv:2507.07969.
>
> [5] Zhang, T., Yu, C., Su, S., & Wang, Y. (2025). ReinFlow: Fine-tuning flow matching policy with online reinforcement learning. arXiv preprint arXiv:2505.22094.
>
> [6] Celik, O., Li, Z., Blessing, D., Li, G., Palenicek, D., Peters, J., ... & Neumann, G. (2025). Dime: Diffusion-based maximum entropy reinforcement learning. arXiv preprint arXiv:2502.02316.
>
> [7] Ren, Allen Z., et al. "Diffusion policy policy optimization." arXiv preprint arXiv:2409.00588 (2024).
>
> [8] Zhang, Tonghe, et al. "ReinFlow: Fine-tuning flow matching policy with online reinforcement learning." arXiv preprint arXiv:2505.22094 (2025).

---

> > ### Comment · Reviewer_rXyd · 2025-11-25
> > **Reply to authors' response**
> >
> > Thank you for your efforts in addressing my comments. I have no further questions at this time, and I have raised my evaluation score accordingly.

---

> > > ### Author Response · Authors · 2025-11-26
> > >
> > > Thank you very much for your follow-up and for raising your score. We are glad that our revisions have satisfactorily addressed your comments. Your constructive feedback has helped us significantly improve the quality of our paper.

---

### Official Review · Reviewer_J2Zm · 2025-11-02

**Soundness:** 2
**Presentation:** 3
**Contribution:** 2
**Rating:** 4
**Confidence:** 4

**Summary:**

The paper extends SAC to flow matching policies. In order to do so, the authors realize that flows can be expressed as residual RNNs, showing the same problems with gradient instabilities. To counter-act this effects, the authors introduce a GRU like flow architecture (FLOW-G) and a transformer like flow architecture (FLOW-T). In ordert to compute the entropy for SAC, the authors show that the flow can be transformed into a stochastic flow using Gaussian noise at each time step and we can take the gradient of the path derivative of the action trajectories to compute the gradient of the entropy. Both approaches are benchmarked on online RL and offline-2-online settings and show competetive performance.

**Strengths:**

- The idea of using time-series architectures to improve the instabilities of flow-matching gradients in RL is novel and promising
- the paper presents the first max-ent framework for flow matching
- benchmarks seem to be competitive, yet I have some doubts regarding some evaluations (see weaknesses)

**Weaknesses:**

- The relation to existing methods is unclear: By introducing the noise augmented rollouts, the approach is very similar to diffusion based methods such as DIME. In DIME, they introduce a lower bound to the entropy which consists of the noise process and reverse noise process path. In this paper, only the reverse noise path is considered fro the gradient, this is the only difference that I can see (except for the architectures). However, while the derivation of using the gradient of the path measure for the entropy seems to be correct, it is unclear why the path measure should be used as additional reward in SAC. This is just done without any further justification. In contrast, the lower bound introduced by DIME provides a much more principled derivation also for the critic update (Showing that the critic update optimizes a clearly defined objective, which is not the case here) while resulting in almost the same algorithm.
- The relation to  DIME needs to discussed in much more detail. As far as I can see it, the paper can be seen as an extension of DIME with more sophisticated architectures (GRU or transformer instead of residual RNN) and the missing forward noise process. However, it is unclear whether this modification benefits or harms the performance.
- I do not believe the reported results for some baselines are correct. DIME is performing much better then the reported results (even though it is evaluated on the benchmark versions v3 instead of v4, but I do not believe that this makes a substantial difference). DIME performs competitive to much stronger baselines than standard SAC such as BRO or SimbaV2, also on harder tasks than presented in this paper. While these baselines use Gaussian policies, they should be added to evaluate if the reported results are really SOTA performance.

**Questions:**

- The authors should investigate why DIME is not performing well for them. Maybe the v4 of the benchmark version requires different hyper-parameters
- Please explain the algorithmic differences to DIME in much more detail (by adding stochasticity to the flow integration steps your process basically becomes a diffussion process, at least the generative process is the same (except for the parametrization)).
- Please also elaborate on why the noise augmented rollout preserves the same marginal densities. This needs further justification or a reference where this has been shown.

---

> ### Author Response · Authors · 2025-11-18
> **Response 1**
>
> We thank the reviewer for this thoughtful comment. We begin by addressing the reviewers' primary concern regarding the distinction between our method and DIME.
>
> >### 1. The relation to existing methods such as DIME. (weakness 1, 2, question 2)
> >> The relation to existing methods is unclear: By introducing the noise augmented rollouts, the approach is very similar to diffusion based methods such as DIME. In DIME, they introduce a lower bound to the entropy which consists of the noise process and reverse noise process path. In this paper, only the reverse noise path is considered fro the gradient, this is the only difference that I can see (except for the architectures). However, while the derivation of using the gradient of the path measure for the entropy seems to be correct, it is unclear why the path measure should be used as additional reward in SAC. This is just done without any further justification. In contrast, the lower bound introduced by DIME provides a much more principled derivation also for the critic update (Showing that the critic update optimizes a clearly defined objective, which is not the case here) while resulting in almost the same algorithm.
>
> >> The relation to DIME needs to discussed in much more detail. As far as I can see it, the paper can be seen as an extension of DIME with more sophisticated architectures (GRU or transformer instead of residual RNN) and the missing forward noise process. However, it is unclear whether this modification benefits or harms the performance.
>
> >> Please explain the algorithmic differences to DIME in much more detail (by adding stochasticity to the flow integration steps your process basically becomes a diffussion process, at least the generative process is the same (except for the parametrization)).
>
>
> We thank the reviewer for this thoughtful and constructive comment. We address the concerns from three complementary perspectives.
>
>
>  #### **(1) On the use of the path measure in the critic and its relation to diffusion methods**
>
> The noise-augmented rollouts indeed naturally bring our formulation closer to diffusion-style methods. The path-measure viewpoint underlying flows and diffusions is well established in Holderrieth & Erives (2025) [1], which we now explicitly reference to clarify this connection. We agree that this does not constitute the core differentiator relative to DIME. Therefore we did not claim it as a main contribution.
>
> We also agree that our original submission did not sufficiently articulate why the joint path density appears inside the SAC critic. To address this, we have added a new subsection in the Appendix, **Path-Regularized Soft Critic**, in which we explicitly decompose the joint rollout density into the marginal action density and the conditional path distribution. This reveals that the surrogate term $-log p_c(A | s)$ corresponds to the standard SAC entropy term supplemented by an additional conditional path-entropy regularization. This provides a principled explanation: the path measure is not intended to replicate the original SAC entropy exactly, but instead acts as a stabilizing regularizer that is particularly helpful for off-policy learning with multi-step flow rollouts.

---

> ### Author Response · Authors · 2025-11-18
> **Response 2**
>
> #### **(2) Differences from DIME in training objectives and training settings**
>
> We respectfully clarify that our method and DIME differ in more than architectural choices. The differences extend not only to the critic and actor objectives, but also to the training setting and applicability:
>
> - **Critic update.**
>   DIME constructs a lower bound on the marginal policy entropy using both forward and reverse diffusion processes, forming a well-defined variational MaxEnt RL objective and training a **distributional Q-function**. Our critic is intentionally kept as a standard SAC soft Q-function (with regularizer), as our goal is to isolate and analyze the impact of stabilizing the actor without introducing additional modeling complexities.
>   As explained in the new Path-Regularized Soft Critic subsection, our critic surrogate corresponds to the SAC entropy plus a conditional path-entropy regularizer, resulting in a distinct training objective compared to DIME.
>
> - **Actor update.**
>   DIME does not optimize the SAC actor objective directly. Instead, it maximizes a surrogate proven to be an upper bound on the SAC objective.
>   In contrast, our actor update follows the exact SAC policy gradient. In the appendix we show that for samples drawn from the rollout:
>   \[
>       \nabla_θ \log π_θ(a|s) = \nabla_θ \log p_c(A|s),
>   \]
>   hence we do not use a surrogate approximation.
>
> - **Training setting: Offline-to-Online vs. from-scratch.**
>   A key practical difference is that **SAC Flow supports Offline-to-Online training**, where the policy is first pretrained using a flow-matching objective and then fine-tuned online. This is crucial for robotic and OGBench-style settings.
>   In contrast, DIME currently supports only from-scratch online MaxEnt RL, and cannot be readily used for Offline-to-Online RL because its learning objective does not decompose into a pretraining stage. For this reason as well, SAC Flow is not an extension or modification of DIME.
>
> Taken together, these differences illustrate that although both methods use noise-augmented paths, the underlying critic, actor objective, and training pipeline diverge fundamentally. Therefore, our method should not be viewed simply as ``DIME with more sophisticated architectures (GRU or transformer instead of residual RNN) and the missing forward noise process.''. Given the distinct solution strategies at this level, the selection of DIME as the baseline is reasonable.
>
> #### **(3) Our main contribution: stabilizing off-policy RL with flow policies (orthogonal to DIME)**
>
> We again appreciate the reviewer’s observation that introducing stochasticity makes our generative process resemble a diffusion process. As clarified in Holderrieth & Erives (2025) [1], noise-augmented flows and diffusion models share the same underlying path formulation; we now cite this explicitly.
>
> However, the primary scope of our paper is different from DIME:
>
> - DIME innovates on **variational MaxEnt RL objectives for diffusion policies** (including a forward-reverse entropy bound and a distributional critic).
> - Our work focuses on **stabilizing off-policy RL** for multi-step flow rollouts, addressing exploding/vanishing gradients documented in Flow Q-learning [3] and FlowRL [4].
>
> Our Flow-G and Flow-T architectures specifically target this instability and can be combined with any MaxEnt RL objective—including DIME's. For example, DIME’s objective could be applied on top of Flow-T/G, potentially reducing its reliance on `stop_gradient` operations.
>
> Regarding whether the architectural changes of Flow-G/T help or harm performance: our ablations (Fig. 5 in the paper) and empirical results across all benchmarks indicate that Flow-G and Flow-T significantly improve training stability and final performance compared to naive flow policy, independent of the MaxEnt RL objective used.
>
> Conversely, SAC Flow is intentionally designed as a simple, direct implementation of SAC for flow policies, rather than as a new MaxEnt RL learning objective. We want to emphasize that our goal is not to propose a new entropy formulation or a new variational MaxEnt RL principle—directions already explored in prior work such as Q-score matching, MaxEntDP [2], and DIME. Instead, our contribution focuses specifically on stabilizing off-policy RL for multi-step flow rollouts through architectural innovations (Flow-G and Flow-T), while keeping the underlying SAC objective (nearly) unchanged. For this reason, our objectives are complementary to the variational MaxEnt RL objectives developed in DIME, rather than competing with.
>
> ---

---

> ### Author Response · Authors · 2025-11-18
> **Response 3**
>
> > ### 2. The performance of DIME (weakness 3, question 1)
> >>I do not believe the reported results for some baselines are correct. DIME is performing much better then the reported results (even though it is evaluated on the benchmark versions v3 instead of v4, but I do not believe that this makes a substantial difference). DIME performs competitive to much stronger baselines than standard SAC such as BRO or SimbaV2, also on harder tasks than presented in this paper. While these baselines use Gaussian policies, they should be added to evaluate if the reported results are really SOTA performance.
>
> >>The authors should investigate why DIME is not performing well for them. Maybe the v4 of the benchmark version requires different hyper-parameters
>
> Thank you for pointing out this issue. We agree that we previously used DIME's **default** parameters, which consequently failed to achieve optimal performance. We have refined the hyperparameters for DIME (detailed in Tab. 4). This has indeed led to an improvement in DIME's performance. We have further (1) standardized the network architecture and the parameter size of DIME to be consistent with all other baselines; (2) used unified hyperparameters across all tasks for fair comparasion.
>
> We have updated the experimental results in Sec. 5. Our SAC Flow achieves comparable or better performance with DIME in Hopper, Walker, HalfCheetah, Ant, and HumanoidStandup, and slightly worse on the Humanoid task. The results that SAC Flow can match or exceed the performance of a strong baseline like DIME serves as strong empirical evidence for the effectiveness of our stabilized actor architectures.
>
> We hope our updated experiments address your concerns.
>
> ---
>
> > ### 3. Why the noise augmented rollout preserves the same marginal densities (question 3)
> >> Please also elaborate on why the noise augmented rollout preserves the same marginal densities. This needs further justification or a reference where this has been shown.
>
> We thank the reviewer for pointing out the need to justify why the noise-augmented rollout preserves the same marginal action densities. This was indeed an oversight in our original submission: although our derivation relies directly on well-known results, we mistakenly omitted the corresponding reference.
>
> We have now explicitly added the correct citation to Holderrieth & Erives (2025) [1] in the revised manuscript. As shown in their detailed derivation (pp. 28–35), introducing noise along the flow trajectory does not alter the marginal distribution of the terminal state; the noise affects only the path measure, not the endpoint distribution. This is precisely the argument we rely on when treating the noise-augmented rollout as producing actions drawn from the same marginal policy.
>
> We apologize for the missing reference in the original submission and thank the reviewer for helping us correct this. The revised version now clearly cites [1] and explains that the formal proof is provided in their analysis.
>
> ---
>
>
> **References:**
>
> [1] Holderrieth, P., & Erives, E. (2025). *An Introduction to Flow Matching and Diffusion Models*.
>
> [2] Dong, X., Cheng, J., & Zhang, X. S. (2025). *Maximum Entropy Reinforcement Learning with Diffusion Policy*.
>
> [3] Park, S., Li, Q., & Levine, S. (2025). *Flow Q-learning*.
>
> [4] Lv, L., Li, Y., Luo, Y., Sun, F., Kong, T., Xu, J., & Ma, X. (2025). *Flow-Based Policy for Online Reinforcement Learning*.

---

> ### Comment · Reviewer_J2Zm · 2025-11-25
>
> Thank you for including the derivation of the path measure for the critic. Thats very informative and improves the paper. Yet, regarding the comparison to DIME, I still have a few questions / recommendations. It is true that there are several differences in terms of algorithmic choices (distributional critic, offline-to-online etc...), but the actor update as well as the critic update are still highly related. I agree that the derivations and intuitions are quite different and the new derivation provides insights, but the core resulting algorithm is, as I believe, quite similar. Thats not necesarily a bad thing, but I think it would be good to ablate exactly this  algorithmic differences. For example, while it is true that DIME is minimizing an upper bound for the actor update instead of the original objective, the gradient of that upper bound is very similar to the gradient presented in this paper. It contains the log densities of the "reverse path", but the gradient of the forward path is the same as far as I can see. The same is true for the critic update. The main difference is the missing "reverse path".
> Both algorithms have different hyper-parameters and it is always hard to say which one has been better tuned. For example, while the distributional critic is known to work better in many algorithms, the min and max ranges of the value function have to be set accordingly in order to get a good performance. To attribute the performance differences to the core algorithmic difference of this paper (include reverse path log densities or not), it would be useful to compare against the DIME update (i.e. include the reverse path log likelilhoods) in exactly the same conditions, i.e., use the same actor and critic architectures.

---

> > ### Author Response · Authors · 2025-11-29
> > **Clarification on DIME Relation, New Ablations, and TD3 Results (1/2)**
> >
> > We sincerely thank the reviewer for the continued engagement and the specific suggestion to ablate the algorithmic differences. These constructive comments pushed us to dive deeper into the DIME implementation and perform additional experiments.
> >
> > Below, we address the reviewer’s comments by (1) clarifying our contribution scope, (2) reporting the requested ablation (DIME Flow-T), and (3) presenting new results on TD3 Flow to demonstrate the generality of our architecture.
> >
> > ### **1. Clarification: Our Core Contribution is Architectural, Not Algorithmic**
> >
> > We would like to reiterate that the primary contribution of this paper is **not** proposing a new reinforcement learning objective. Instead, our focus is on **stabilizing off-policy RL training for Flow-based policies** by treating the flow generation process as a sequential modeling problem.
> >
> > We chose SAC simply as a representative, standard off-policy algorithm to demonstrate that our proposed **Flow-T (Transformer)** and **Flow-G (GRU)** architectures can solve the gradient instability issues (vanishing/exploding gradients) inherent in multi-step flow rollouts.
> >
> > * **Why we call it SAC Flow:** We strictly implement the standard SAC algorithm. As noted in our previous response, the shift from marginal entropy to path entropy is merely the mathematical consequence of applying SAC's principle to flow policies.
> > * **Differentiation:** We respectfully submit that our method should not be viewed as an extension of DIME. DIME introduces a specific variational lower bound involving forward/reverse processes. Our method is simply "SAC applied to Flow," where the stability comes from the architecture, not a new variational objective.
> >
> > ### **2. Ablation Study: SAC Flow-T vs. DIME Flow-T (Addressing the Specific Request)**
> >
> > To directly address the reviewer’s hypothesis that the performance gains might stem from the DIME objective rather than our architecture (or vice versa), we followed your recommendation to compare them under identical conditions.
> >
> > **Implementation Details:**
> > We carefully examined the DIME paper and codebase. We identified that DIME involves two key components absent in SAC Flow:
> > 1.  **Forward Process Prior:** DIME injects a prior gradient $\nabla \log p_{prior}(x)$ during the forward sampling (noise injection).
> > 2.  **Reverse Path Loss:** DIME includes the reverse path log-likelihood in the update, calculated via the probability ratio with $p_{prior}$.
> >
> > We implemented a new variant, **DIME Flow-T**, which applies the full DIME objective (including the forward prior injection and reverse path loss) on top of our Flow-T architecture. This implementation has been updated in our supplementary code (`dime_continuous_action_multi_steps_transformer_velocity_jax.py`).
> >
> > **Results:**
> > We compared **SAC Flow-T** (our method) against **DIME Flow-T** using the same hyperparameters (inherited from SAC Flow-T for fairness, with DIME-specific priors kept as default). The results are added in the Appendix F.5 and shown in Figure 24. We got these two key observations:
> >
> > * **Observation 1:** DIME Flow-T performs significantly better than the original DIME (where a standard diffusion policy is used), confirming that our Flow-T architecture indeed contributes significantly to performance and stability.
> > * **Observation 2:** However, **SAC Flow-T still outperforms (or matches) DIME Flow-T** across the benchmark tasks.
> >
> > **Conclusion:** This ablation suggests that while the DIME objective is theoretically elegant, the additional complexity (reverse path, prior injection) does not necessarily yield better performance than the simpler SAC objective when a sufficiently stable architecture (Flow-T) is used. This reinforces our claim that stabilizing the architecture is the bottleneck for off-policy Flow RL, and SAC Flow provides a simpler, effective solution.

---

> > ### Author Response · Authors · 2025-11-29
> > **Clarification on DIME Relation, New Ablations, and TD3 Results (2/2)**
> >
> > ### **3. Generalization: Stabilizing TD3 with Flow-T (New Experiment)**
> >
> > To strictly prove that our contribution is orthogonal to DIME and lies in the stabilization of generic off-policy algorithms, we extended our evaluation to **TD3**, an algorithm that completely lacks an entropy term (and thus has no relation to the path-entropy/DIME debate). The results are shown in Appendix F.4.
> >
> > We implemented **TD3 Flow**, replacing the standard MLP actor in TD3 with our Flow-T and Flow-G architectures.
> >
> > * **Results:** As shown in the updated **Figure 23**, TD3 Flow-T and TD3 Flow-G converge stably and achieve high performance.
> > * **Implication:** Since TD3 does not maximize entropy (marginal or path-based), the success of TD3 Flow cannot be attributed to any "DIME-like" objective.
> >
> > This result serves as strong empirical evidence that our contribution is a **universal stabilizer for off-policy flow policies**, applicable to both maximum-entropy methods (SAC) and deterministic policy gradients (TD3).
> >
> > ### **Summary**
> >
> > We have updated the manuscript to include:
> > * The **DIME Flow-T ablation**, discussing the trade-offs between the DIME objective and the simpler SAC objective under the same architecture.
> > * The **TD3 Flow results**, demonstrating the universality of the Flow-T/G architectures.
> >
> > We hope these rigorous comparisons clarify that our work offers a distinct, architecture-centric contribution that simplifies and stabilizes off-policy RL for flows, independent of the DIME formulation. We kindly ask the reviewer to reconsider the assessment based on these new evidences.

---

### Public Comment · ~Haque_Ishfaq1 · 2025-11-15
**Comparing against Langevin Soft Actor Critic  from ICLR 2025**

Dear authors,

This seems like a really nice work. We wanted to point out that our ICLR 2025 paper on Langevin Soft Actor Critic [1] should be used as a baseline. In LSAC, we use Langevin Monte Carlo based Thompson Sampling and distributional critic to achieve sample efficiency.

In our work, we used `DSAC-T` (Duan et al., 2023), `DIPO` (Yang et al., 2023), `SAC` (Haarnoja et al., 2018a), `TD3` (Fujimoto et al., 2018), `PPO` (Schulman et al., 2017), `TRPO` (Schulman et al., 2015), `REDQ` (Chen et al., 2021) and `QSM` (Psenka et al., 2024) as baselines and we got superior results than these baselines in MuJoCo tasks and DeepMind Control Suite tasks. We see, you also used `QSM`, `SAC` and `PPO` as baseline. Another method that you used as baseline, `DIME`, was published in ICML 2025, after our work was published.

Comparing our plot from Figure 1 in [1] and your Figure 4, it seems LSAC will be a stronger baseline than the baselines you used. Although we used MuJoCo `v-3` whereas yours used `v-4`, however we believe the difference would be minor if you run `LSAC` in `v-4`. Our codebase and all the data are publicly available here https://github.com/hmishfaq/LSAC

We would appreciate if you could provide a comparison against LSAC in your experiments.

Thanks!

[1] Ishfaq, Haque, et al. "Langevin Soft Actor-Critic: Efficient Exploration through Uncertainty-Driven Critic Learning."  ICLR 2025

---

> ### Author Response · Authors · 2025-11-18
> **Response**
>
> Dear authors of LSAC,
>
> Thank you for sharing your work, Langevin Soft Actor-Critic [1], and for your interest in our paper. We appreciate the advancements you have made regarding uncertainty-driven critic learning and efficient exploration.
>
> Regarding the baseline comparison, we would like to clarify the specific technical scope and motivation of our work:
>
> **Focus on Flow Policy Optimization via SAC:**
>
> Our primary contribution is addressing the optimization instability—specifically the Backpropagation Through Time (BPTT) gradient explosion—that arises when training multi-step flow-based policies with off-policy algorithms. Our proposed architectures (Flow-G and Flow-T) are designed specifically to resolve this stability bottleneck inherent to ODE-based actors.
>
> **Context of Sample Efficiency and Orthogonality:**
>
> Our discussion of "sample efficiency" primarily refers to the inherent advantage of enabling off-policy training for flow models, which is significantly more data-efficient than the standard on-policy approaches (like PPO) commonly used in this domain.
>
> Crucially, our work is orthogonal to research focused on general sample-efficient RL algorithms (such as LSAC). Your work innovates on the Critic and Exploration mechanisms, whereas our work focuses on stabilizing the Actor architecture for generative models. We are not proposing a new general-purpose sample-efficient RL algorithm, but rather solving the specific optimization challenge to make flow-based policies trainable with standard SAC.
>
> Given this distinction, we focused our comparison on other generative policy baselines (e.g., DIME, FlowRL, QC-FQL).
>
>
>
> Best regards,
>
> The Authors

---

### Author Response · Authors · 2025-11-18
**General Response**

We sincerely appreciate the reviewers' constructive feedback. We have carefully refined the paper, supplemented the experiments, and provided point-by-point responses to all questions. Specifically, our revisions include:

(1) Following the reviewers' suggestions, we have refined the main results and ablation studies to better evaluate the performance of SAC Flow.

(2) We expanded the discussion in Sections 2 and 3 regarding multimodal action distributions and exploding/vanishing gradients.

(3) We added two new sections to the Appendix: A.5 Path-Regularized Soft Critic, and B. Detailed Analysis of Gradient Stability.

(4) We included extensive additional experiments in Appendix F, analyzing baseline performance and the sensitivity of SAC Flow to the specifics of GRUs and Transformers.

(5) We further reported the IQM return and Probability of Improvement metrics in Appendix F.

(6) We revised the color schemes in Figure 4 and Figure 7 to enhance the clarity of the experimental results.

We hope that our revisions and responses have satisfactorily addressed your concerns. We look forward to answering any further questions you may have.

The revised paper has been uploaded, with all changes highlighted in blue. Our detailed responses to specific questions are provided below.

---

### Author Response · Authors · 2025-11-29
**Rebuttal Summary: Positive Consensus (Scores: 8, 8, 6) & Distinction from DIME (1/2)**

Dear Area Chair,

We thank the reviewers for their constructive feedback and the time invested in reviewing our work. We are encouraged by the positive consensus, with **Reviewer 4 (dSSA) maintaining Accept (8)**, **Reviewer 2 (rXyd) raising their score to Accept (8)**, and **Reviewer 3 (bZhq) raising their score to Weak Accept (6)**.

Below is a summary of how we addressed the main concerns and the new experiments added during the rebuttal.

### 1. Addressed Concerns on Empirical Evidence & Instability (Reviewer 3, bZhq, Score raised: 4 $\to$ 6)
**Timeline Note:** We wish to highlight that **prior to the recent platform incident**, Reviewer 3 explicitly confirmed that concerns 2, 3, and 4 were fully resolved, isolating the "gradient instability evidence" (Concern 1) as the *sole remaining issue*. The subsequent score raise directly followed our provision of the requested revision.

* **Our Response:** We significantly revised Section 5.3 and **replaced Figure 6** to show an aligned analysis of **Episodic Return** vs. **Gradient Norms** side-by-side. Furthermore, we introduced a "Gradient Ratio" analysis across three environments (Ant, Walker, Hopper).
* **Key Result:** We demonstrated that the Naive Flow baseline consistently exhibits a **~10x relative explosion** in gradient norms (compared to our stabilized method) at the exact moment learning collapses.
* **Outcome:** Reviewer 3 found the new aligned plots and unified analysis convincing, stating their concerns were resolved, and **raised their score to 6**.

### 2. Addressed Concerns on Mathematical Justification (Reviewer 2, rXyd, Score raised: 6 $\to$ 8)
Reviewer 2’s main concern was that the proposed architectures (Flow-G/T) were presented as heuristic choices without sufficient mathematical motivation regarding the instability.

* **Our Response:** We added **Appendix B.1 and B.2**, providing a formal derivation of the Jacobian product explosion in multi-step flow rollouts (linking to BPTT instability). We theoretically justified how the gating mechanism (in Flow-G) and Pre-LN/Residuals (in Flow-T) act as structural stabilizers for the Jacobian product.
* **Outcome:** The reviewer found the mathematical analysis "informative" and the arguments well-supported, **raising their score to 8**.

---

> ### Author Response · Authors · 2025-11-29
> **Rebuttal Summary: Positive Consensus (Scores: 8, 8, 6) & Distinction from DIME (2/2)**
>
> ### 3. Addressed Concerns on the Distinction from DIME (Reviewer 1, J2Zm)
> Reviewer 1’s primary concern was the relationship between our method and DIME. We addressed this through a multi-step clarification and experimental verification process:
>
> * **Initial Clarification:** We first clarified that our method differs from DIME in three key aspects: **Critic** (Standard Soft Q vs. Distributional), **Actor** (Direct Gradient vs. Surrogate Objective), and **Training Setting** (Supports Offline-to-Online vs. Online-only).
> * **Reviewer’s Follow-up:** The reviewer acknowledged our derivation of the path measure was *"very informative"* but recommended an ablation study to isolate the effect of the algorithmic objective from the architecture to definitively prove our contribution.
> * **Our Execution (New Experiments):**
>     1.  **DIME Objective Ablation:** We implemented **DIME Flow-T**, applying the full DIME objective (including forward priors and reverse path likelihoods) on top of our Flow-T architecture (Appendix F.5).
>         * Result: **SAC Flow-T matches or outperforms DIME Flow-T** (Fig. 24). This confirms that the performance gains stem from our **architectural stabilization**, and that the simpler SAC objective is sufficient when the architecture is stable.
>     2.  **Generalization to Deterministic RL:** To prove orthogonality to entropy-based formulations (like DIME), we applied our architecture to **TD3** (Appendix F.4).
>         * Result: **TD3 Flow converges stably** (Fig. 23). Since TD3 has no entropy/path-measure term, this conclusively proves our contribution is a **universal stabilizer**, independent of the MaxEnt/DIME formulation.
>
> * **Summary of Differences:** Through these clarifications and experiments, we have demonstrated that SAC Flow and DIME are fundamentally distinct in their core contributions:
>     * **Architectural vs. Objective-Centric:** Our work focuses on network structural modifications (Flow-G/T) to inherently stabilize off-policy training, viewing the flow as a sequential model. In contrast, DIME focuses on proposing a specific MaxEnt RL objective tailored for diffusion policies (involving variational lower bounds).
>     * **Orthogonality:** As a result, our architectural contribution is general and orthogonal to the choice of objective—it can stabilize standard SAC, TD3 (as shown in Appendix F.4), or even the DIME objective itself (as shown in Appendix F.5).
>     * **Functionally:** Our approach also supports **Offline-to-Online** fine-tuning and **Deterministic** policy gradients, capabilities not covered by the DIME paper.
>
> ### Conclusion & Contribution
> In summary, this paper offers a novel **Sequential Model Perspective** on flow-based policies. By formalizing the flow rollout as being algebraically equivalent to a residual RNN, we identify **gradient instability** as the root cause of training failures.
>
> * **Innovation:** We propose **Flow-G** and **Flow-T**, which reparameterize the velocity network using modern sequential modeling principles to resolve these gradient pathologies.
> * **Universality:** We demonstrated that our solution is **orthogonal** to learning objectives (effective with SAC, TD3, and DIME objectives).
> * **Performance:** We achieved SOTA performance on both dense-reward (MuJoCo) and sparse-reward (OGBench/Robomimic) benchmarks.
>
> We thank the AC for considering our response and the improved consensus among reviewers.

---

### Meta-Review · Area_Chair_dxaa · 2026-01-07

**Summary:**

There are concerns around Empirical Evidence & Instability, Mathematical Justification and Distinction from DIME. Authors did a great job addressing all the concerns.

**Reviewer Concerns:**

Concerns on Empirical Evidence & Instability and concerns on Mathematical Justification are both addressed and reviewers acknowledged. Concerns on the Distinction from DIME is also addressed although reviewer did not respond at the end.

**Reviewer Scores:**

Reviewer 3, bZhq, Score raised: 4 -> 6
Reviewer 2, rXyd, Score raised: 6 -> 8

---

### Decision · Program_Chairs · 2026-01-26

Accept (Poster)